# Random features and polynomial rules

Fabián Aguirre-López[1,2,3⋆], Silvio Franz[1,4] and Mauro Pastore[1,5,6†]

**1** Université Paris-Saclay, CNRS, LPTMS, 91405 Orsay, France
**2** LadHyX, UMR, CNRS 7646, École polytechnique, 91128 Palaiseau, France
**3** Chair of Econophysics and Complex Systems, École polytechnique, 91128 Palaiseau, France
**4** Dipartimento di Matematica e Fisica, Università del Salento, 73100 Lecce, Italy
**5** Laboratoire de physique de l'École normale supérieure, CNRS, PSL University, Sorbonne University, Université Paris-Cité, 24 rue Lhomond, 75005 Paris, France
**6** The Abdus Salam International Centre for Theoretical Physics, Strada Costiera 11, 34151 Trieste, Italy

⋆ fabian.aguirre-lopez@polytechnique.edu , † mpastore@ictp.it

## Abstract

Random features models play a distinguished role in the theory of deep learning, describing the behavior of neural networks close to their infinite-width limit. In this work, we present a thorough analysis of the generalization performance of random features models for generic supervised learning problems with Gaussian data. Our approach, built with tools from the statistical mechanics of disordered systems, maps the random features model to an equivalent polynomial model, and allows us to plot average generalization curves as functions of the two main control parameters of the problem: the number of random features $N$ and the size $P$ of the training set, both assumed to scale as powers in the input dimension $D$. Our results extend the case of proportional scaling between $N$, $P$ and $D$. They are in accordance with rigorous bounds known for certain particular learning tasks and are in quantitative agreement with numerical experiments performed over many order of magnitudes of $N$ and $P$. We find good agreement also far from the asymptotic limits where $D \to \infty$ and at least one between $P/D^K$, $N/D^L$ remains finite.

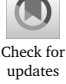 Check for updates

# 1 Introduction

The connection between deep feed-forward neural networks (DNNs) in the large-width limit and kernel methods has been well understood in the last years. It has been shown, in a Bayesian learning perspective, that if the number of units in each hidden layer is taken to infinity at fixed input dimension and training set size, a DNN becomes a "neural network Gaussian process" whose kernels can be defined iteratively layer by layer [1–4]. This result has been recently generalized beyond the infinite-width limit [5–10]. In a dynamical perspective moreover, it has been shown that wide DNNs trained with gradient-based methods exhibit the lazy-training kernel regime [11], evaluated by first order Taylor-expanding the network with respect to the weights around initialization [12–14].

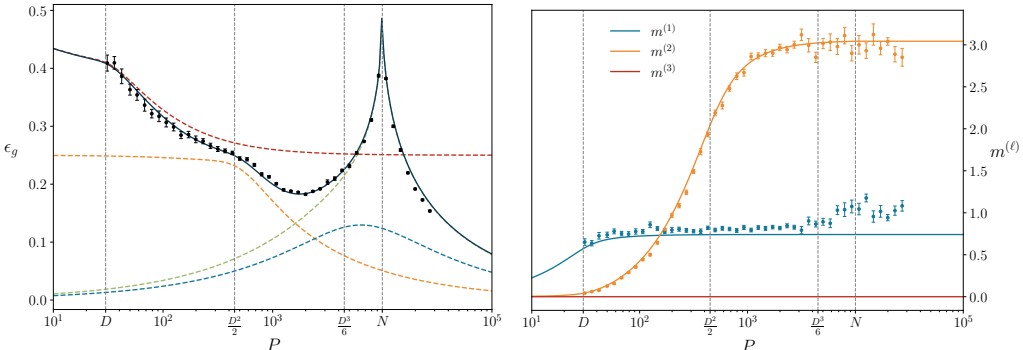

Figure 1: **Left:** generalization error of the RFM on a classification task, as a function of the size of the training set $P$, for $D = 30$, $N = 10^4$, weights regularization $\zeta = 10^{-8}$, *quadratic teacher* (balanced: $\tau_1 = \tau_2 = 1/\sqrt{2}$, $\tau_{\ell > 2} = 0$) and ELU activation functions (defined in Eq. (8) below); the continuous line is the equivalent polynomial theory devised in Sec. 4, truncated at $L = 3$; dashed lines are the asymptotic theories (see Sec 6 for details) for $N \to \infty$ and $P/D$ finite (red), $N \to \infty$ and $P/\binom{D}{2}$ finite (yellow), $N \to \infty$ and $P/\binom{D}{3}$ finite (blue), $P/\binom{D}{3}$ and $N/P$ finite (green); black points are results from numerical experiments averaged over 50 instances (see Appendix I). The model learns the linear features (first step at $P \sim O(D)$), then learns the quadratic features (second step at $P \sim O(D^2)$), then follows the interpolation peak at $P \sim N$. **Right:** numerical and theoretical teacher-student overlaps – defined in Eq. (37) and (45) – of the linear and quadratic features (the overlap of the cubic features is identically 0 by definition); the parameters of the model are the same as for the left panel.

Once a DNN is proven equivalent to a kernel machine, the mechanism by which it realizes the input-output mapping of the corresponding supervised-learning task is understood: the input data, which generally speaking are points in $\mathbb{R}^D$, are mapped with an implicit *feature map* $\psi : \mathbb{R}^D \to \mathbb{R}^N$ to an $N$-dimensional space where the classification, or regression, rule is linear and can be learnt by the read-out layer. The mapping to the feature space is implicit, in the sense that the learning problem can be solved by a support vector machine (SVM), so that learning and generalization depend on the features only through the kernel $\bar{\mathcal{H}}(\mathbf{x}, \mathbf{x}') = \sum_{i=1}^{N} \psi_i(\mathbf{x}) \psi_i(\mathbf{x}')/N$ (see, for reference, [15]). Learning curves (generalization error as a function of the size $P$ of the training set) of kernel machines can be obtained analytically from a statistical mechanics [16–19] or a mathematical [20–22] perspective. A very interesting trait of these curves is their staircase shape for $P \sim D^K$: by setting the scaling of the size of the training set to a certain power $K$ of the input dimension, features of order $K$ can be learnt by the machine, so that the test error decreases increasing $K$ with subsequent steps.

The discovery of the lazy training regime of wide neural networks motivated in the recent past the study of the *random features model* (RFM) [23, 24], a shallow (one-hidden-layer, 1HL) neural network where the feature map is explicitly parametrized by a fixed random linear embedding of the input points from $\mathbb{R}^D$ to $\mathbb{R}^N$, followed by a non-linear activation function. In this sense, the model mimics the behavior of a neural network in the large-width limit, where the feature map depends only on initialization and learning is linear.

In the present work we study theoretically the generalization performance of the RFM in the large-$D$ limit for empirical risk minimization, with $P \sim D^K$, $N \sim D^L$. We find, under a quite general teacher/student setting with a random polynomial teacher and Gaussian i.i.d. input data, that

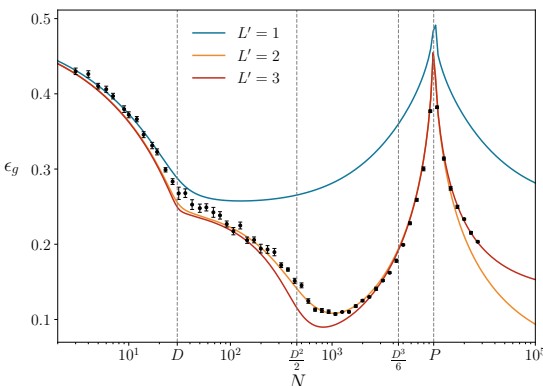

Figure 2: Generalization error of a RFM on a classification task, as a function of the number of hidden units $N$, for $P = 10^4$ and the rest of the parameters as in Fig. 1; continuous lines are the theories truncated at $L' = 1, 2, 3$ (respectively: blue, yellow, red); numerical points (in black) are nicely interpolating between these curves in the regimes where $N \sim O(D), O(D^2), O(D^3)$, validating Eq. (25), where the truncation $L'$ of the equivalent polynomial theory is fixed at $L \sim \log(N)/\log(D)$.

- as long as $P \ll N$, the model behaves as an infinite-rank ($N \to \infty$) kernel machine: for $P \sim D^K$, features of order $K$ can be learnt, such that the generalization error as a function of $P$ has a staircase descent (or overfitting peaks if the teacher is less complex) with steps corresponding to different values of $K$;

- for $P \gg N$ and $N \sim D^L$, the model is equivalent to a degree-$L$ polynomial student: if the complexity of the teacher is lower than the degree $L$, the generalization error is equal to zero, or otherwise, to the minimum error for a degree-$L$ polynomial fitting a more complex teacher;

- for $P \sim N$, an interpolation peak of the generalization error, which depends on the strength of the regularization of the student's weights, occurs.

This behavior is depicted in Fig. 1. Comparison with numerical experiments shows that our theory, based on the mapping of the RFM to an *equivalent noisy polynomial model*, predicts well the quantitative behavior of the true generalization performance at finite size, over many orders of magnitude.

Our theory, formulated from the point of view of the statistical mechanics of disordered systems, expresses the generalization performance of the RFM in terms of few order parameters with a clear physical interpretation, as overlaps between combinations of the student's weights and the parameters defining the teacher. In this way, we are offering a complementary take on what is known about RFMs in the computer science community, as we discuss in the following.

## 1.1 Related works

In this section we give an overview on the previous works that have been of inspiration to our paper, presenting relevant results and differences with our approach.

Random feature models were introduced in [23–26], initially as randomized low-rank approximations of kernels arising in classification or regression problems. Recently, their interest was renewed by the discovery that DNNs behaves as RFMs close to the infinite-width limit, both in a Bayesian learning [1–4] and in a gradient-based learning [11–14] setting. This mapping, which provides one of the few limits where DNNs can be studied with analytical methods, has

motivated in the last few years a huge effort to formalize their behavior in terms of expressive power and generalization performance.

In particular, the impressive series of works [14,27–33] (see [34] for a review) formulates rigorous bounds on the generalization performance of RFMs in different asymptotic regimes. For a non-exhaustive recap of the results (with our notation):

- In [27], the large-$D$ limits where $D^{L+\delta} \leq N \leq D^{L+1-\delta}$ (for small $\delta$) after sending $P \to \infty$ (underparametrized regime) and $D^{K+\delta} \leq P \leq D^{K+1-\delta}$ after sending $N \to \infty$ (overparametrized regime) are considered. In the first case the model is found equivalent to degree-$L$ polynomial regression; in the second one, it reduces to (infinite-rank) kernel regression, which for that number of samples can fit at most a degree-$K$ polynomial in the inputs, in a way also investigated in literature [16–22].

- In [29], the limit where both $N$ and $P$ scale linearly with $D$ with their ratio fixed is considered; the generalization error as a function of the ratio between the number of hidden units and the size of the training set first decreases for $N/P$ small, then exhibits a peak at the interpolation threshold $N/P = 1$ and then relaxes again for $N \gg P$ to the value predicted from the kernel theory with $P \sim D$, coherently with the previous point. This phenomenology is widely observed in numerical experiments and known in literature as *double descent* [35] of the generalization error.

- In [31], the authors push forward the analysis of [27] (that is, $P$ and $N$ scaling polynomially with $D$) to the regimes where $N \leq P^{1-\delta}$ and $N \geq P^{1+\delta}$. The authors show indeed that the limiting behavior is given by the smallest of $N$ and $P$, and they find the interpolation threshold at $N \sim P$ also in this polynomial scaling.

- In [33], universality results on training and test error are proven in the $P \sim N$ regime for a larger class of models, as long as with finite dimensional outputs, and generic losses. Indeed, they prove that training and test errors depend on the random features distribution only through its covariance structure.

These papers find bounds to the generalization performance of a RFM with rigorous analytical methods under quite general assumptions on data distribution and activation functions.

A statistical mechanics point of view, complementary to the formal approach discussed so far, has been formulated in the series of papers [36–42]. Originally aiming at modelling the role of data structure in machine learning, as in other contemporary approaches [43–50], the authors obtained in [37] a closed-form expression for the generalization error of RFMs for regression and classification in the asymptotic regime where $N \sim P \sim D$. Their approach, based on the replica theory from statistical mechanics [51], can be applied to supervised learning tasks with generic convex loss functions. Not only their results are supported under mild hypothesis by analytical proofs [29,33,38,52,53], but they can predict remarkably well the numerical experiments. Our work extends these results to more general scaling regimes, where $P \sim D^K$, $N \sim D^L$.

One of the main steps in our derivation is the expansion of activation function of the hidden layer on a polynomial basis, which corresponds to the diagonalization of the kernel (20) on its eigenbasis (Mercer's decomposition). This expansion is then truncated to a certain degree $L$, corresponding to the integer exponent in the scaling law $N \sim D^L$: similar approximations appeared recently in [54, 55]. Moreover, while the literature on the double descent behavior of the generalization error is vast and impossible to outline here (see for example [35]), we mention [56], where the presence of more than one peak in the generalization curve is remarked: the authors call "linear peak" the one occurring at $P \sim D$ for $N \gg P$, where the model behaves as a kernel learning the linear features, while for $P \sim N$ there is a "non-linear

Table 1: Notations used in this paper.

| Symbol | Definition |
|---|---|
| $D$ | input space dimension |
| $N \sim D^L$ | feature space dimension |
| $P \sim D^K$ | size of the training set |
| $B$ | degree of the teacher |
| $n$ | number of replicas |
| $\eta_\ell$ | $N / \binom{D}{\ell}$ |
| $\alpha, \beta, \cdots$ | indices in input space |
| $i, j, \cdots$ | indices in feature space |
| $\mu, \nu, \cdots$ | indices spanning the training set |
| $a, b, \cdots$ | indices in replica space |
| $\boldsymbol{\alpha}$ | multi-index $\{\alpha_1, \cdots, \alpha_\ell\}$, $\alpha_1 < \cdots < \alpha_\ell$ |
| $\theta$ | teacher parameters, $\theta = \{\theta_{\boldsymbol{\alpha}}^{(\ell)}\}_{\ell=1}^{B}$ |
| $F$ | $N \times D$ random features matrix |
| $\mathbf{F}_{\alpha}, \mathbf{F}_i$ | $(F_{i\alpha})_{i=1}^{N}$, $(F_{i\alpha})_{\alpha=1}^{D}$ |
| $\mathbf{F}_{\boldsymbol{\alpha}}^{\otimes \ell}$ | $(F_{i\alpha_1} \cdots F_{i\alpha_\ell})_{i=1}^{N}$ |
| $C$ | $FF^\top / D$ |
| $C^{\odot \ell}$ | $((C_{ij})^\ell)_{i,j=1}^{N} \simeq \sum_{\boldsymbol{\alpha}} \mathbf{F}_{\boldsymbol{\alpha}}^{\otimes \ell} (\mathbf{F}_{\boldsymbol{\alpha}}^{\otimes \ell})^\top / \binom{D}{\ell}$ |
| $Q, Q^{(\ell)}, \dots$ | $(Q_{ab})_{a,b=1}^{n}$, $(Q_{ab}^{(\ell)})_{a,b=1}^{n}$, $\dots$ |

peak" due to the non-linearity of the activation function acting as noise and overfitted when $P$ and $N$ are of the same order; in the present work we show that, as long as $N \gg P$, there is a peak (or a descent) for each of the regimes $P \sim D^K$.

Appeared in parallel with our work, the paper [57] pushes forward the line of research of [29] from a mathematical perspective, deriving sharp asymptotics for the generalization of random features ridge regression in the polynomial regime. The even more recent [58] bounds the test error of random features ridge regression with a dimension-free (that is, for arbitrary input dimension $D$) non-asymptotic (depending explicitly on $N$ and $P$, converging to the test error when at least one of them is large) deterministic equivalent, depending only on the feature map eigenvalues through a set of self-consistent equations. The mapping of our approach to [57, 58] is left for future work.

## 2 The model

We would like to study the generalization performance of the Random Features model in a teacher/student [59, 60] supervised learning set-up, where the teacher performs an input-output mapping with various degree of complexity. We summarize in Table 1 the main notations used in this paper.

The input data $\mathbf{x}$ are vectors in $\mathbb{R}^D$ with i.i.d. Gaussian elements, while the labels are assigned by a polynomial teacher of degree $B$ defined as:

$$
\begin{aligned}
y &\sim p(y \,|\, v(\mathbf{x})) \,, \\
v(\mathbf{x}) &= \sum_{\ell=1}^{B} \frac{\tau_\ell}{\sqrt{\binom{D}{\ell}}} \sum_{\alpha_1 < \cdots < \alpha_\ell} \theta_{\alpha_1 \cdots \alpha_\ell}^{(\ell)} x_{\alpha_1} \cdots x_{\alpha_\ell} \,,
\end{aligned}
\tag{1}
$$

where $\theta_{\alpha}^{(1)}$, $\theta_{\alpha\beta}^{(2)}$, $\cdots$ are i.i.d. $\mathcal{N}(0,1)$ parameters collectively denoted as $\theta$, describing the

non-linear decision boundary (diagonal terms, irrelevant for large $D$, are for simplicity not included in the sum). Notice that the function $v(\mathbf{x})$ coincide with the Hamiltonian of the "mixed $p$-spin model" of the statistical physics of the spin-glasses (see, for example, [61]). The mixture parameters $\tau_\ell$, weighting the monomials of different degree, are chosen to respect $\sum_{\ell=1}^{B} \tau_\ell^2 = 1$. Within this general setting, we will concentrate on the specific simple examples of a deterministic teacher for binary classification or a noisy teacher for polynomial regression with variance of the noise $\Delta$, for which Eq. (1) reduces respectively to

$$y \sim \delta\left[y - \mathrm{sgn}\, v(\mathbf{x})\right], \qquad y \sim \mathcal{N}\left[y \,|\, v(\mathbf{x}), \Delta\right]. \tag{2}$$

It has been shown in [16] that a *polynomial* student, defined in the same way as in Eq. (1), would learn the weights of the teacher in a hierarchical fashion: $O(D^K)$ examples are needed in order to learn the parameters $\theta^{(\ell)}$ for $\ell \leq K$. However, here the student's task is to learn the weights of the last layer of a 2-layers NN, $f(\mathbf{x}; \mathbf{w})$, whose first layer realizes a random embedding of the data in a $N$-dimensional feature space:

$$f(\mathbf{x}; \mathbf{w}) = \phi\left[\lambda(\mathbf{x}; \mathbf{w})\right], \tag{3}$$

$$\lambda(\mathbf{x}; \mathbf{w}) = \frac{1}{\sqrt{N}} \sum_{i=1}^{N} w_i \, \sigma\left(\frac{1}{\sqrt{D}} \sum_{\alpha=1}^{D} F_{i\alpha} x_\alpha\right), \tag{4}$$

where $F$ is a $N \times D$ quenched random matrix with i.i.d. standard normal entries, $\sigma$ is the non-linear activation function of the hidden layer, $\mathbf{w} \in \mathbb{R}^N$ the student's weight vector and $\phi$ the activation function of the last ("readout") layer. It is customary to introduce the pre-activations

$$h_i = \frac{1}{\sqrt{D}} \sum_{\alpha=1}^{D} F_{i\alpha} x_\alpha, \tag{5}$$

which at fixed instance of the random features $F$, given that we chose $x_\alpha$ i.i.d normal variables, follow a multivariate Gaussian distribution with covariance

$$C_{ij} = \mathbb{E}_{\mathbf{x}^\mu}[h_i h_j] = \frac{1}{D} \sum_{\alpha=1}^{D} F_{i\alpha} F_{j\alpha}. \tag{6}$$

In our setting with independent random features, $C$ is a Wishart matrix.

While our theory is general in the choice of $\sigma$ (as long as it can be expanded on the basis of Hermite polynomials – see Sec. 4), we will test our results for popular choices, such as

$$\sigma(h) = \mathrm{ReLU}(h) = \max(h, 0), \tag{7}$$

$$\sigma(h) = \mathrm{ELU}(h) = \begin{cases} \exp(h) - 1, & \text{if } h < 0, \\ h, & \text{if } h \geq 0, \end{cases} \tag{8}$$

(respectively, Rectified and Exponential Linear Unit).

The training set is made of $P$ input-output pairs, $\mathcal{T} = \{(\mathbf{x}^\mu, y^\mu)\}_{\mu=1}^{P}$. The student learns by solving the following optimization problem,

$$\mathbf{w}^\star = \underset{\mathbf{w}}{\mathrm{argmin}} \left[\sum_{\mu=1}^{P} \mathcal{L}[y^\mu, \lambda(\mathbf{x}^\mu; \mathbf{w})] + \frac{\zeta}{2} \|\mathbf{w}\|^2\right], \tag{9}$$

where $\mathcal{L}$ is an opportune convex loss function and $\zeta$ controls the regularization of the weights. Notice how the solution of this optimization problem is an implicit function of the training set

$\mathcal{T}$, the parameters of the teacher $\theta$ and the random features $F$, that is $\mathbf{w}^\star = \mathbf{w}^\star(\mathcal{T}, \theta, F)$; we will omit this dependence to lighten notations.

The choice of the loss function $\mathcal{L}$ and the readout activation function $\phi$ in Eq. (3) defines the specific learning task to perform. The approach we present in the following can be followed for any choice of $\mathcal{L}$, as long as the optimization problem (9) is convex (to justify the Replica Symmetric ansatz, see below); this is true in particular if $\mathcal{L}(y, \lambda)$ is convex as a function of $\lambda$, as the student's weights $\mathbf{w}$ enter linearly in the definition of $\lambda(\mathbf{x}; \mathbf{w})$. However, to simplify formulas, we will report in the main text only the case of a pure quadratic loss, reading, both in the case of regression and classification:

$$\mathcal{L}(y, \lambda) = \frac{1}{2}(y - \lambda)^2 \,. \tag{10}$$

The use of a regression loss for a classification task ($\lambda$ instead of $\phi(\lambda)$ even when $\phi = \mathrm{sgn}$) is not unusual in practical cases (e.g. the `linear_model.RidgeClassifier` class in the Scikit-learn library for Python [62]) and dates back to the early days of NNs [60, 63].

The main aim of this work is the evaluation of the generalization performance of the model, both for the classification and the regression problems, using a statistical mechanics approach. From this perspective, the model defines a disordered system with $N$ degrees of freedom $\mathbf{w}$, and quenched disorder given by the realization of the input points $\mathbf{x}^\mu$, the teacher's parameters $\theta$ and the random features $F$. Our computation will follow the standard path, starting from the partition function at inverse temperature $\beta$

$$\mathcal{Z} = \int \mathrm{d}\mathbf{w} \exp\left[-\beta \sum_{\mu=1}^{P} \mathcal{L}[y^\mu, \lambda(\mathbf{x}^\mu; \mathbf{w})] - \frac{\beta\zeta}{2}\|\mathbf{w}\|^2\right] \,. \tag{11}$$

## 3 Generalization error

In order to quantify how well the student can learn the teacher, we look at the generalization error, defined as the probability of misclassifying a new sample (in the case of classification) or as the mean squared error of a new point (in the case of regression). Given a test point $(\mathbf{x}, y) \sim p_0(\mathbf{x})p(y|\nu(\mathbf{x}))$, both cases can be expressed with the following formula,

$$\epsilon_g(\mathcal{T}, \theta, F) = \int \mathrm{d}\mathbf{x}\, p_0(\mathbf{x}) \int \mathrm{d}y\, p(y|\nu(\mathbf{x})) \frac{1}{4^\kappa} \left[y - \phi(\lambda(\mathbf{x}; \mathbf{w}^\star))\right]^2 \,, \tag{12}$$

where $\kappa = 1$ for binary classification and $\kappa = 0$ for regression. Notice the presence of the function $\phi$ in the definition of the generalization error, at variance with the loss function (10).

With (12) we can evaluate the quality of the student NN (3) for a given realization of the teacher, of the random weights $F$, and of the dataset $\mathcal{T}$. In order to get a general view of the effectiveness of (3), we calculate the average generalization error over all the sources of randomness. Doing so, we get a function of $N$, $P$, and $D$ only,

$$\epsilon_g = \int \mathrm{d}\nu\, \mathrm{d}\lambda\, p(\nu, \lambda) \int \mathrm{d}y\, p(y|\nu) \frac{1}{4^\kappa} \left[y - \phi(\lambda)\right]^2 \,,$$

$$p(\nu, \lambda) = \mathbb{E} \int \mathrm{d}\mathbf{x}\, p_0(\mathbf{x}) \delta(\nu - \nu(\mathbf{x})) \delta(\lambda - \lambda(\mathbf{x}; \mathbf{w}^\star)), \tag{13}$$

where we took $\mathbb{E} = \mathbb{E}_{\mathcal{T}, \theta, F}$.

We have written the average generalization error as in Eq. (13) to show that we only need to know the joint distribution of $(\nu, \lambda)$ to evaluate it. Since $\mathbf{x}$ is a test point, and is

thus uncorrelated with $\mathbf{w}^\star$, we will take the distribution $p(v,\lambda)$ as Gaussian: to compute the generalization error we only need the first and second moments,

$$
\begin{aligned}
0 &= \mathbb{E}[v], & t^\star &= \mathbb{E}[\lambda], \\
\rho &= \mathbb{E}[v^2], & m^\star &= \mathbb{E}[v\lambda], & q^\star &= \mathbb{E}[\lambda^2] - t^{\star 2}.
\end{aligned}
\tag{14}
$$

Notice that by definition of the model (*i.e.* the normalization of the mixing parameters $\tau_\ell$) $\rho$ is identically equal to 1. In section 5 we will show how to obtain these quantities from a replica approach. Stating formally hypotheses on $\mathbf{w}^\star$, $F$ and the functions $v(\mathbf{x})$, $\lambda(\mathbf{x};\mathbf{w})$ in order to justify this ansatz is beyond the scope of this paper: we will check *a posteriori* its validity with numerical experiments. Central limit theorems for sums of non linear functions of Gaussian fields (the pre-activations (5) at given feature matrix $F$), of the kind we just used to motivate this ansatz, have been proven in the past under rather technical conditions on the realization of the feature-feature covariance matrix $C$ and of the vector $\mathbf{w}^\star$ [33,38,53,64,65]. The interested reader can find a sketch of proof in [36], Appendix A.2, where the moments of the variables $\lambda$ are evaluated and the leading order diagrams identified as the Gaussian ones.

For the case of binary classification with $y = \mathrm{sgn}(v)$ and $\phi = \mathrm{sgn}$,

$$
\begin{aligned}
\epsilon_g &= \frac{1}{4}\mathbb{E}\left([y - \mathrm{sgn}(\lambda)]^2\right) \\
&= \int_{-\infty}^{0} Dv\left[1 - H\left(\frac{t^\star + m^\star v}{\sqrt{q^\star - m^{\star 2}}}\right)\right] + \int_{0}^{\infty} Dv\, H\left(\frac{t^\star + m^\star v}{\sqrt{q^\star - m^{\star 2}}}\right),
\end{aligned}
\tag{15}
$$

where we use the Gardner notation [66] $Dv = \frac{e^{-v^2/2}}{\sqrt{2\pi}}\,dv$ and $H(x) = \int_x^\infty Dt$. Notice that when $t^\star = 0$ (that is, when the student is zero-mean) the formula simplifies to

$$
\epsilon_g = \frac{1}{\pi}\arccos\left(\frac{m^\star}{\sqrt{q^\star}}\right).
\tag{16}
$$

For the case of noisy polynomial regression, ($\phi = \mathrm{id}$ and $\Delta = \mathbb{E}[(y-v)^2]$) [67,68],

$$
\epsilon_g = \mathbb{E}[(y-\lambda)^2] = \rho + \Delta - 2m^\star + q^\star + t^{\star 2}.
\tag{17}
$$

These formulas remind the generalization error of a generalized linear model with the same architecture as the teacher [60]: in that case, $m^\star/\sqrt{q^\star}$ corresponds to the angle between the teacher and the student weight vectors. For the RFM, it is not clear *a priori* if we can interpret $m^\star/\sqrt{q^\star}$ as a scalar product of the teacher's weight vector and some effective weights of the student. If this can be done, the RFM could be mapped to an equivalent polynomial model. In Sec. 4 we will show how to explicitly construct it from $\mathbf{w}$ and $F$, thus achieving this mapping. To do so, we need to spend a few words on the connection between RFMs and kernel machines, in order to explain the truncation of the activation function $\sigma$ on the basis of Hermite polynomials, which we will use later on.

## 4 Kernel learning and polynomial models

The RFM defined in (3) is a generalized linear model in the learnable parameters $\mathbf{w}$, so it can be formulated as a kernel model, as we remind in this section. First of all, for the particular choice of quadratic loss, we can write down the explicit solution to (9),

$$
w_i^\star = \frac{1}{\sqrt{N}}\sum_j\left(\zeta\mathbb{1}_N + \frac{P}{N}\bar{\mathcal{K}}\right)_{ij}^{-1}\sum_\mu y^\mu\sigma(h_j^\mu),
\tag{18}
$$

where the pre-activations $h$ are given by (5) and the operator

$$\bar{\mathcal{K}}_{ij} = \frac{1}{P} \sum_{\mu} \sigma(h_i^{\mu}) \sigma(h_j^{\mu}), \qquad (19)$$

defines the kernel in feature space. The properties of the kernel are crucial for the generalization performances.

While our analysis will be more general, in this section we consider the limit $P \to \infty$, for the purpose of arguing.[1] In this case the empirical kernel reduces to

$$\mathcal{K}_{ij} = \mathbb{E}_{\mathbf{x}^{\mu}}[\sigma(h_i^{\mu}) \sigma(h_j^{\mu})]. \qquad (20)$$

From this formula, it is possible to obtain an explicit formula of the kernel $\mathcal{K}$ as a function of the covariance matrix of the pre-activations (6). To this aim, as the pre-activations are Gaussian, it is convenient to expand the activation function on the basis of Hermite polynomials (see also [27]):

$$\sigma(h_i) = \sum_{\ell=0}^{\infty} \frac{\mu_{\ell}}{\ell!} \mathrm{He}_{\ell}(h_i), \qquad (21)$$

where $\mathrm{He}_{\ell}$ is the $\ell$-th Hermite polynomial and the coefficient $\mu_{\ell}$ are:

$$\mu_{\ell} = \int Dx \, \mathrm{He}_{\ell}(x) \sigma(x). \qquad (22)$$

Along these lines, the kernel (20) can be expressed for large $D$ [69,70] (see App. A for details) as

$$\mathcal{K}_{ij} = \sum_{\ell=0}^{\infty} \frac{\mu_{\ell}^2}{\ell!} (C_{ij})^{\ell}, \qquad (23)$$

where $C_{ij}$, given by (6), is a rank-$D$ Wishart matrix with elements $C_{ii} = 1 + O(D^{-1/2})$ and $C_{ij} = O(D^{-1/2})$ for $i \neq j$. The matrix with entries $(C_{ij})^{\ell}$, which we denote by $C^{\odot \ell}$, defines an interesting random matrix ensemble, obtained taking Hadamard (element by element) powers of the covariance $C$. A similar ensemble was recently studied in [71].

Suppose now the relation between $N$ and $D$ is fixed: $N \sim D^{L+\delta}$ with $0 \leq \delta < 1$. The $N \times N$ matrix $C^{\odot \ell}$ has generically rank equal to $\min\{D^{\ell}/\ell!, N\}$ (neglecting possible smaller contributions to the rank coming from outliers, see Sec. 6 where we discuss more in detail the properties of these matrices) and off-diagonal elements $O(D^{-\ell/2})$. For $\ell > L$ the matrix is full ranked, the small off-diagonal terms give a vanishing contribution to eigenvalues and eigenvectors. In other words, when $D^{\ell}$ is scaling faster than $N$ to infinity, we can take the large $D$ limit *before* the large $N$ one in the combination

$$(C_{ij})^{\ell} = \delta_{ij}[1 + \ell O(D^{-1/2})] + (1 - \delta_{ij}) O(D^{-\ell/2}) \underset{D \text{ large}, \, D^{\ell} \gg N}{\simeq} \delta_{ij}, \qquad (24)$$

in the same way as the Wishart matrix $C_{ij} = \delta_{ij}[1 + O(D^{-1/2})] + (1 - \delta_{ij}) O(D^{-1/2})$ concentrates around $\delta_{ij}$ for $D \gg N$ (the Marchenko-Pastur distribution, providing the asymptotic distribution of the spectrum of $C$, concentrates around 1 for $N/D \to 0$). We can thus truncate the expansion substituting $C^{\odot \ell > L}$ by the identity matrix:

$$\mathcal{K}_{ij} \simeq \sum_{\ell=0}^{L} \frac{\mu_{\ell}^2}{\ell!} (C_{ij})^{\ell} + \mu_{\perp, L}^2 \delta_{ij}, \qquad (25)$$

---

[1]We do so in this section to introduce the kernel $\mathcal{K}$ as a limit of the empirical kernel $\bar{\mathcal{K}}$; in the replica approach in Sec. 5 the expectation over the data will be taken explicitly and the kernel will appear naturally without taking the $P \to \infty$ limit from the start.

where

$$\mu_{\perp,L}^2 = \sum_{\ell=L+1}^{\infty} \frac{\mu_\ell^2}{\ell!} = \mathbb{E}_x[\sigma(x)^2] - \sum_{\ell=0}^{L} \frac{\mu_\ell^2}{\ell!}, \tag{26}$$

for $x \sim \mathcal{N}(0,1)$.

This truncation is proven for $L = 1$ (that is, in the proportional regime $N \sim D$) in [72], and extended to the case $L > 1$ under generic assumptions on the kernel $\mathcal{K}$ in [31,55]. A convincing check of this property for moderately large values of $N$ is given by Fig. 2, which shows the theoretical curves of the generalization error obtained through a truncated effective theory (that we describe below) at different values of $L'$, compared with the numerical experiments, as a function of $N$; quantitative agreement is obtained for $L' = L \sim \log N / \log D$, with the numerical points interpolating nicely the theoretical curves in the various regimes.

The analysis above suggests that in the $N \sim D^L$ regime we can represent the RFM as an effective noisy polynomial student

$$\lambda_{\mathrm{eff}}(\mathbf{x}^\mu; \mathbf{w}) = \mu_0 m^{(0)} + \sum_{\ell=1}^{L} \frac{\mu_\ell}{\sqrt{D^\ell}} \sum_{\alpha_1,\cdots,\alpha_\ell} s_{\alpha_1\cdots\alpha_\ell}^{(\ell)} : x_{\alpha_1}^\mu \cdots x_{\alpha_\ell}^\mu : + z^\mu, \tag{27}$$

where

- $m^{(0)} = \sum_i w_i / \sqrt{N}$ is the empirical mean of the vector $\mathbf{w}$, rescaled by $\sqrt{N}$;

- the student parameters $s_{\alpha_1\cdots\alpha_\ell}^{(\ell)}$ are the scalar product of $\mathbf{w}$ with the "vectors" $\mathbf{F}_{\alpha_1\cdots\alpha_\ell}^{\otimes\ell} / \sqrt{N}$ with components $F_{i\alpha_1} \cdots F_{i\alpha_\ell} / \sqrt{N}$ (see Table 1),

$$s_{\alpha_1\cdots\alpha_\ell}^{(\ell)} = \frac{1}{\sqrt{N}} \sum_i w_i F_{i\alpha_1} \cdots F_{i\alpha_\ell}; \tag{28}$$

- we have written the expansion of the Hermite polynomials in terms of the so-called Wick products of the $x$'s, routinely used in theoretical physics and defined from the following generating function (see for example [73]):

$$: x_1 \cdots x_k := \partial_{\lambda_1} \cdots \partial_{\lambda_k} G(\boldsymbol{\lambda}; \mathbf{x})\big|_{\boldsymbol{\lambda}=0}\,,$$
$$G(\boldsymbol{\lambda}; \mathbf{x}) = \frac{\exp(\boldsymbol{\lambda}^\top \mathbf{x})}{\mathbb{E}[\exp(\boldsymbol{\lambda}^\top \mathbf{x})]} = \exp(\boldsymbol{\lambda}^\top \mathbf{x} - \|\boldsymbol{\lambda}\|^2/2)\,. \tag{29}$$

These quantities have the property $\mathbb{E}[: x_1 \cdots x_k :] = 0$. The mapping

$$\mathrm{He}_\ell(h_i) \simeq \sum_{\alpha_1,\cdots,\alpha_\ell} \frac{F_{i\alpha_1} \cdots F_{i\alpha_\ell}}{\sqrt{D^\ell}} : x_{\alpha_1} \cdots x_{\alpha_\ell} :, \tag{30}$$

which is true for $D$ large and which we used to write $\lambda_{\mathrm{eff}}$ in terms of Wick products starting from (21), is proven in App. B;

- the last term $z^\mu$ is a Gaussian noise term with zero mean and variance $\mathbb{E}(z^{\mu 2}(\mathbf{w})) = \mu_{\perp,L}^2 \sum_{i=1}^{N} w_i^2 / N$ which can be represented as

$$z^\mu = \frac{\mu_{\perp,L}}{\sqrt{N}} \sum_{i=1}^{N} w_i v_i^\mu, \tag{31}$$

in terms of i.i.d. $\mathcal{N}(0,1)$ variables $v_i^\mu$.

Although ultimately the parameters $\mathbf{s}^{(\ell)}$ and $\mathbf{z}$ are functions on the network weights, to enlighten the notation we will not explicitly write the dependence on $\mathbf{w}$.

In (27) we give an effective description of the RFM, mapping it to a polynomial model with correlated weights in presence of a noise term coming from the $\ell > L$ terms in the expansion (21). The mapping is motivated by the fact that $\lambda_{\text{eff}}(\mathbf{x}^\mu; \mathbf{w})$ defined in this way, admits as second moment $\mathbf{w}^\top \mathcal{K} \mathbf{w}/N$ at given $F$ and $\mathbf{w}$, with the kernel truncated according to (25); we show this explicitly for the replicated version of $\lambda$ in Appendix C, together with the covariance structure with the polynomial $v(\mathbf{x})$ defining the teacher. This is an extension to generic scaling regimes $N \sim D^L$ of the *Gaussian equivalence principle* from [38] and related works, to which it reduces when $L = 1$. In the following, we will base our analysis on this representation of $\lambda$. This description makes more transparent the meaning of the observables introduced in Sec. 3 and the mechanism by which the RFM learns the teacher's features, as we explain in the following.

# 5 Replica calculation

Let us now turn to the analysis of the general case through the replica method. To obtain the generalization error we write the joint probability distribution of $v$ and $\lambda$ in Eq. (13) as the zero temperature limit of the equilibrium distribution of a statistical mechanics system, as

$$p(v, \lambda) = \lim_{\beta \to \infty} \mathbb{E} \int d\mathbf{w} \frac{1}{\mathcal{Z}} e^{-\beta \sum_\mu \mathcal{L}[y^\mu, \lambda(\mathbf{x}^\mu; \mathbf{w})] - \frac{\beta\zeta}{2} \|\mathbf{w}\|^2} \int d\mathbf{x}\, p_0(\mathbf{x}) \delta(v - v(\mathbf{x})) \delta(\lambda - \lambda(\mathbf{x}; \mathbf{w})). \quad (32)$$

Through a standard application of the replica trick we rewrite the distribution as

$$p(v, \lambda) = \lim_{n \to 0} \lim_{\beta \to \infty} \mathbb{E} \int \prod_{a=1}^n d\mathbf{w}^a e^{-\beta \sum_{\mu,a} \mathcal{L}[y^\mu, \lambda(\mathbf{x}^\mu; \mathbf{w}^a)] - \frac{\beta\zeta}{2} \sum_a \|\mathbf{w}^a\|^2}$$

$$\times \int d\mathbf{x}\, p_0(\mathbf{x}) \delta(v - v(\mathbf{x})) \delta(\lambda - \lambda(\mathbf{x}; \mathbf{w}^1)), \quad (33)$$

which can be obtained from the calculation of the $n$-times replicated partition function

$$Z_n = \mathbb{E}[\mathcal{Z}^n] = \int \prod_{a=1}^n d\mathbf{w}^a\, e^{-\frac{\beta\zeta}{2} \sum_a \|\mathbf{w}^a\|^2} \mathbb{E}_{F,\theta} \left[ \mathbb{E}_{v,\{\lambda^a\}} \int dy\, p(y|v) e^{-\beta \sum_a \mathcal{L}(y, \lambda^a)} \right]^P. \quad (34)$$

In this integral, we treat the distribution of $v$ and $\lambda^a$ conditioned by $F$, $\theta$ and $\mathbf{w}^a$ as Gaussian, with moments given by

$$t_a = \mathbb{E}(\lambda_a | F, \theta), \qquad M_a = \mathbb{E}(v \lambda_a | F, \theta), \qquad Q_{ab} = \mathbb{E}(\lambda_a \lambda_b | F, \theta) - t_a t_b. \quad (35)$$

from which we can extract the generalization error according to (15), (17). Using the representation (27) we can decompose these order parameters as (see Appendix C for details)

$$t_a = \mu_0 M_a^{(0)}, \qquad M_a = \sum_{\ell=1}^{\min\{L,B\}} \frac{\mu_\ell \tau_\ell}{\sqrt{\ell!}} M_a^{(\ell)}, \qquad Q_{ab} = \mu_{\perp,L}^2 Q_{ab}^{(0)} + \sum_{\ell=1}^L \frac{\mu_\ell^2}{\ell!} Q_{ab}^{(\ell)}, \quad (36)$$

with the definitions:

$$M_a^{(0)} = \frac{1}{\sqrt{N}} \sum_{i=1}^N w_i^a, \qquad M_a^{(\ell)} = \frac{\boldsymbol{\theta}^{(\ell)} \cdot \mathbf{s}_a^{(\ell)}}{\binom{D}{\ell}}, \qquad Q_{ab}^{(0)} = \frac{1}{N} \sum_{i=1}^N w_i^a w_i^b, \qquad Q_{ab}^{(\ell)} = \frac{1}{N} \sum_{i,j=1}^N w_i^a C_{ij}^\ell w_j^b, \quad (37)$$

where we are using the notation

$$\boldsymbol{\theta}^{(\ell)} \cdot \mathbf{s}_a^{(\ell)} = \sum_{\boldsymbol{\alpha}} \theta_{\boldsymbol{\alpha}}^{(\ell)} s_{a,\boldsymbol{\alpha}}^{(\ell)} \tag{38}$$

(remember that the sum over $\boldsymbol{\alpha}$ is restricted to ordered tuples).

Enforcing these definition with delta functions in Fourier representation, and anticipating saddle point integration for the various $M$ and $Q$, and their Fourier conjugated parameters that we denote as $\hat{M}$ and $\hat{Q}$ with the due indices, we rewrite the partition function as

$$
\begin{aligned}
Z_n = {} & e^{P S_P[Q,M]} e^{\frac{N}{2}\sum_{a,b}\hat{Q}_{ab}^{(0)}Q_{ab}^{(0)} + \frac{1}{2}\sum_{\ell,a,b}\binom{D}{\ell}\hat{Q}_{ab}^{(\ell)}Q_{ab}^{(\ell)} + \sum_{\ell,a}\binom{D}{\ell}\hat{M}_a^{(\ell)}M_a^{(\ell)}} \\
& \times \mathbb{E}_{F,\theta} \int d\mathbf{w}\, e^{-\frac{1}{2}\mathbf{w}^{\top}\left[(\beta\zeta\mathbb{1}_n+\hat{Q}^{(0)})\otimes\mathbb{1}_N + \sum_{\ell}\hat{Q}^{(\ell)}\otimes\frac{C^{\odot\ell}}{\eta_{\ell}}\right]\mathbf{w} - \sum_{\ell,i,a,\boldsymbol{\alpha}}\hat{M}_a^{(\ell)}w_i^a F_{i,\boldsymbol{\alpha}}^{\otimes\ell}\theta_{\boldsymbol{\alpha}}^{(\ell)}/\sqrt{\eta_{\ell}\binom{D}{\ell}}},
\end{aligned}
\tag{39}
$$

where now $\mathbf{w} \in \mathbb{R}^{n \times N}$, the sums over $\ell$ span $\{1, \cdots, L\}$, $\eta_{\ell} = N/\binom{D}{\ell}$ and

$$S_P[Q,M] = \log \mathbb{E}_{\nu,\{\lambda^a\}} \int dy\, p(y|\nu) e^{-\beta \sum_a \mathcal{L}(y,\lambda^a)}. \tag{40}$$

In writing Eq. (39), we took $\hat{M}_a^{(0)} \to 0$, as the Fourier conjugate of the mean $t_a$ is suppressed in the large-$N$ limit [66] (a property that could be checked *a posteriori* from the saddle point equation for $\hat{M}_a^{(0)}$);[2] moreover, the conventional scalings with $N$ and $\binom{D}{\ell}$ in this equation are chosen in such a way that the hat variables corresponding to the asymptotic regimes explained in Sec. 6 have a non-trivial high-dimensional limit.

Averaging over $\theta$ we obtain:[3]

$$
\begin{aligned}
Z_n = {} & e^{P S_P[Q,M]} e^{\frac{N}{2}\sum_{a,b}\hat{Q}_{ab}^{(0)}Q_{ab}^{(0)} + \frac{1}{2}\sum_{\ell,a,b}\binom{D}{\ell}\hat{Q}_{ab}^{(\ell)}Q_{ab}^{(\ell)} + \sum_{\ell,a}\binom{D}{\ell}\hat{M}_a^{(\ell)}M_a^{(\ell)}} \\
& \times \mathbb{E}_{F} \int d\mathbf{w}\, e^{-\frac{1}{2}\mathbf{w}^{\top}\left[(\beta\zeta\mathbb{1}_n+\hat{Q}^{(0)})\otimes\mathbb{1}_N + \sum_{\ell}(\hat{Q}^{(\ell)}-\hat{M}^{(\ell)}\hat{M}^{(\ell)\top})\otimes\frac{C^{\odot\ell}}{\eta_{\ell}}\right]\mathbf{w}},
\end{aligned}
\tag{41}
$$

and integrating over $\mathbf{w}$,

$$Z_n = e^{P S_P[Q,M]} e^{\frac{N}{2}\sum_{\ell,a,b}\hat{Q}_{ab}^{(0)}Q_{ab}^{(0)} + \frac{1}{2}\sum_{\ell,a,b}\binom{D}{\ell}\hat{Q}_{ab}^{(\ell)}Q_{ab}^{(\ell)} + \sum_{\ell,a}\binom{D}{\ell}\hat{M}_a^{(\ell)}M_a^{(\ell)} - \frac{1}{2}\operatorname{Tr}\log\left[A^{(0)}\otimes\mathbb{1}_N + \sum_{\ell}B^{(\ell)}\otimes C^{\odot\ell}\right]}, \tag{42}$$

where traces are taken over replica and feature indices and we introduced for compactness the $n \times n$ matrices

$$A^{(0)} = \beta\zeta\mathbb{1}_n + \hat{Q}^{(0)}, \qquad B^{(\ell)} = (\hat{Q}^{(\ell)} - \hat{M}^{(\ell)}\hat{M}^{(\ell)\top})/\eta_{\ell}. \tag{43}$$

We notice at this point that, given $N \sim D^{L+\delta}$, for $\ell \leq L$ the matrices $C^{\odot\ell}$ have rank $r_{\ell} = O(D^{\ell}) \ll N$ and have eigenvalues of order $N/\binom{D}{\ell}$. Simple perturbation theory shows that adding these matrices with coefficients of order $1$ only slightly modify the eigenvalues. This is due to the fact that the row spaces (that is, the complements to their null spaces) corresponding to the different $\ell$ are almost orthogonal (we postpone a throughout discussion

---

[2]The terms depending on $\hat{M}^{(0)}$ are given by

$$S_{\hat{M}^{(0)}} = \frac{M^{(0)\top}\hat{M}^{(0)}}{\sqrt{N}} + \frac{1}{2}\hat{M}^{(0)\top}\frac{1}{N}\sum_{i,j}\left[(\beta\zeta\mathbb{1}_n+\hat{Q}^{(0)})\otimes\mathbb{1}_N + \sum_{\ell}(\hat{Q}^{(\ell)}-\hat{M}^{(\ell)}\hat{M}^{(\ell)\top})\otimes\frac{C^{\odot\ell}}{\eta_{\ell}}\right]_{ij}^{-1}\hat{M}^{(0)},$$

so that the saddle point equation for $\hat{M}^{(0)}$ gives $\hat{M}^{(0)} = O(1/\sqrt{N})$.

[3]For the sake of simplicity, to write Eq. (41) we collected a common $C^{\odot\ell}$ between the terms $\hat{Q}^{(\ell)}$ and $\hat{M}^{(\ell)}\hat{M}^{(\ell)\top}$, even though the average over the teacher gives instead a term $\sum_{\boldsymbol{\alpha}} \mathbf{F}_{\boldsymbol{\alpha}}^{\otimes\ell}(\mathbf{F}_{\boldsymbol{\alpha}}^{\otimes\ell})^{\top}/\binom{D}{\ell}$, with ordered indices $\alpha$'s, in front of $\hat{M}^{(\ell)}\hat{M}^{(\ell)\top}$. See discussion around Eq. (49).

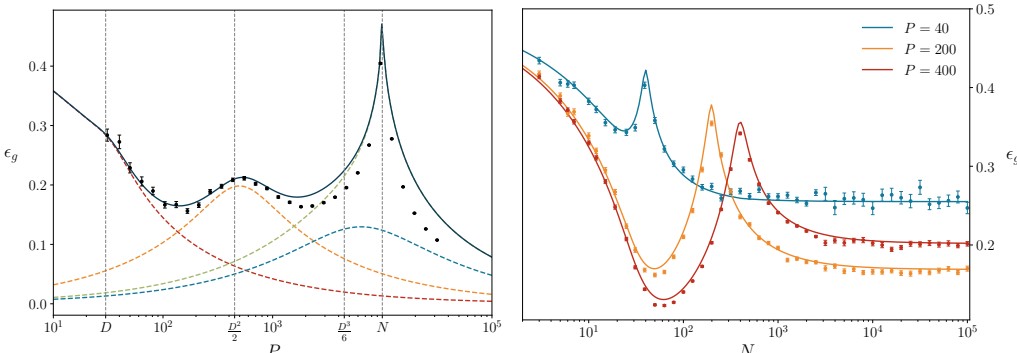

Figure 3: **Left**: generalization error of the RFM on a classification task, as a function of the size of the training set $P$, for $D = 30$, $N = 10^4$, weights regularization $\zeta = 10^{-8}$, *linear teacher* ($\tau_1 = 1$, $\tau_{\ell>1} = 0$) and ELU activation functions; the continuous line is the mean-field theory truncated at $L = 3$; dashed lines are the asymptotic theories for $P/D$ finite and $L > 1$ (red), $P/\binom{D}{2}$ finite and $L > 2$ (yellow), $P/\binom{D}{3}$ finite and $L > 3$ (blue), $P/\binom{D}{3}$ finite and $L = 3$ (green); black points are results from numerical experiments averaged over 50 instances (see Appendix I). The model learns the linear features (first step at $P \sim O(D)$), then overfits the quadratic features before learning they are zero (peak at $P \sim O(D^2)$), then follows the interpolation peak $P \sim N$. Notice how the accordance between the mean-field theory and the experiment is only qualitative around the last peak. **Right**: Generalization error on classification for a linear teacher, as a function of the number of random features $N$, for different amounts of data $P$ ($D = 30$, $\zeta = 10^{-4}$, see Appendix I). The optimal amount of hidden units, for which $\epsilon_g$ is minimal, shifts from overparametrization to underparametrization, as it is visible in the curves for $P = 40$ and $P = 200, 400$. At fixed value of $N$, not always more data means better generalization: after the interpolation peak, the order between the red ($P = 400$) and yellow ($P = 200$) curves is reversed (point of view complementary to the plot in the left panel, where, at fixed $N$, the error can increase with $P$). The curves as functions of $N$ are obtained by gluing together the theories truncated at the corresponding $L$.

on this point to Sec. 6, where we collect and motivate the assumptions we are using on the matrices $C^{\odot \ell}$). In such a situation we approximate the trace-log term appearing in (42) as

$$\text{Tr} \log \left[ A^{(0)} \otimes \mathbb{1}_N + \sum_{\ell=1}^{L} B^{(\ell)} \otimes C^{\odot \ell} \right] \simeq N(1-L) \text{Tr} \log(A^{(0)}) + \sum_{\ell=1}^{L} \text{Tr} \log \left( A^{(0)} \otimes \mathbb{1}_N + B^{(\ell)} \otimes C^{\odot \ell} \right) \quad (44)$$

(notice that Tr in $\text{Tr} \log(A^{(0)})$ is over replica indices only). We report a detailed derivation of Eq. (44) under the hypothesis of orthogonality of the $C^{\odot \ell}$ row spaces in Appendix E. Notice that we could have gotten to the same result decomposing the vectors $\mathbf{w}$ on the row spaces of the $C^{\odot \ell}$ supposed orthogonal. This decomposition clearly shows the hierarchical nature of learning.

## 5.1 Replica symmetric theory

In order to complete the evaluation of the partition function, we need to specify the form of the replica parameters. In this paper we use the replica symmetry (RS) ansatz

$$Q_{ab}^{(\ell)} = \frac{\chi^{(\ell)}}{\beta} \delta_{ab} + q^{(\ell)}, \qquad M_a^{(\ell)} = m^{(\ell)}, \qquad t_a = t. \quad (45)$$

Notice that the diagonal elements of the matrix $Q^{(\ell)}$ are $Q_{aa}^{(\ell)} = \frac{\chi^{(\ell)}}{\beta} + q^{(\ell)}$. We anticipate the scaling with $\beta$ of the variables $\chi$: the quantities $Q_{aa}^{(\ell)} - q^{(\ell)}$ measures the variance of the variables $\lambda$, tending to zero for $\beta \to \infty$. This implies the following form for the conjugate order parameters in the RS:

$$\hat{Q}_{ab}^{(\ell)} = \beta \hat{\chi}^{(\ell)} \delta_{ab} - \beta^2 \hat{q}^{(\ell)}, \qquad \hat{M}_a^{(\ell)} = -\beta \hat{m}^{(\ell)}. \tag{46}$$

Exploiting the explicit parametrization of the RS matrices, we can perform the traces over replica indices in Eq. (44), to get (see Appendix F)

$$\mathrm{Tr} \log \left( A \otimes \mathbb{1}_N + B \otimes C^{\odot \ell} \right) = nN \log(\beta \hat{\chi}^{(\ell)}) + n \, \mathrm{Tr} \log(\gamma_\ell \mathbb{1} + C^{\odot \ell}) - n \beta \eta_\ell \frac{\hat{q}^{(0)}}{\hat{\chi}^{(\ell)}} \mathrm{Tr}(\gamma_\ell \mathbb{1} + C^{\odot \ell})^{-1}$$

$$- n \beta \frac{\hat{q}^{(\ell)} + (\hat{m}^{(\ell)})^2}{\hat{\chi}^{(\ell)}} \mathrm{Tr}[C^{\odot \ell} (\gamma_\ell \mathbb{1} + C^{\odot \ell})^{-1}], \tag{47}$$

where we introduced the parameter

$$\gamma_\ell = \eta_\ell \frac{(\zeta + \hat{\chi}^{(0)})}{\hat{\chi}^{(\ell)}}, \tag{48}$$

and the remaining traces are over feature indices only.

We need now to evaluate the traces in feature indices. In order to proceed, we make at this point a crucial approximation, and treat $C^{\odot \ell}$ as a matrix with a Merchenko-Pastur spectrum with parameter $\eta_\ell = N/\binom{D}{\ell}$. This amounts essentially in approximating $C^{\odot \ell}$, by

$$C_{ij}^{\odot \ell} = \frac{\ell!}{D^\ell} \sum_{\alpha_1 < ... < \alpha_l} F_{\alpha_1}^i F_{\alpha_1}^j ... F_{\alpha_\ell}^i F_{\alpha_\ell}^j, \tag{49}$$

i.e. in neglecting the terms with equal indices $\alpha$ in the sum that defines $C^{\odot \ell}$. While this approximation can be fully justified in the regimes where $N, D \to \infty$ with $N/D^L$ finite, as we will see, it turns out to be an excellent approximation even for moderately large values of the parameters (see Sec. 6 and Appendix D for an extended discussion on this point).

Using the properties of the resolvent of large random matrices (see Appendix D), we can write that, for large $N$,

$$\frac{1}{N} \mathrm{Tr}(\gamma_\ell \mathbb{1} + C^{\odot \ell})^{-1} \approx g_\ell(-\gamma_\ell), \tag{50}$$

where $g_\ell$ is the Stieltjes transformation of the Marchenko-Pastur distribution with ratio $\eta_\ell = N/\binom{D}{\ell}$:

$$g_\ell(z) = \frac{1 - z - \eta_\ell - \sqrt{(1 - z - \eta_\ell)^2 - 4z\eta_\ell}}{2z\eta_\ell}. \tag{51}$$

Re-arranging terms we get, for large $\beta$,

$$Z_n \sim e^{PS_P + NS_M}, \tag{52}$$

where

$$\frac{1}{\beta n} S_M = -\sum_{\ell=1}^{\min\{L,B\}} \frac{m^{(\ell)} \hat{m}^{(\ell)}}{\eta_\ell} + \frac{1}{2} \sum_{\ell=0}^{L} \frac{q^{(\ell)} \hat{\chi}^{(\ell)} - \chi^{(\ell)} \hat{q}^{(\ell)}}{\eta_\ell} + \frac{(1-L)}{2} \frac{\hat{q}^{(0)}}{\zeta + \hat{\chi}^{(0)}}$$

$$+ \frac{1}{2} \sum_{\ell=1}^{L} \eta_\ell \frac{\hat{q}^{(0)}}{\hat{\chi}^{(\ell)}} g_\ell(-\gamma_\ell) + \frac{1}{2} \sum_{\ell=1}^{L} \frac{\hat{q}^{(\ell)} + (\hat{m}^{(\ell)})^2}{\hat{\chi}^{(\ell)}} [1 - \gamma_\ell g(-\gamma_\ell)], \tag{53}$$

and for the quadratic loss (10),

$$\frac{1}{\beta n} S_P = \frac{2m^\star \langle y\, v \rangle - q^\star - \langle (t^\star - y)^2 \rangle}{2(1 + \chi^\star)}\,, \tag{54}$$

where $\langle \cdot \rangle = \int \mathrm{d}y\, D\, v\, p(y|v)(\cdot)$ is the average over the teacher distribution (1) and

$$
\begin{aligned}
m^\star &= \sum_{\ell=1}^{\min\{L,B\}} \frac{\tau_\ell \mu_\ell}{\sqrt{\ell!}} m^{(\ell)}\,, & t^\star &= \mu_0 m^{(0)}\,, \\
\chi^\star &= \mu_\perp^2 \chi^{(0)} + \sum_{\ell=1}^{L} \frac{\mu_\ell^2}{\ell!} \chi^{(\ell)}\,, & q^\star &= \mu_\perp^2 q^{(0)} + \sum_{\ell=1}^{L} \frac{\mu_\ell^2}{\ell!} q^{(\ell)}\,.
\end{aligned}
\tag{55}
$$

A detailed derivation of the terms $S_M$ and $S_P$, with the form of $S_P$ valid for generic loss functions, is reported in Appendix G.

Eq. (55) gives the RS version of Eq. (36): these quantities are precisely the ones appearing in Eq. (14), giving the low-order statistics of the distribution used to evaluate the generalization error. Once their value is known from the saddle point equations implicit in the derivation of the partition function, they can be used to obtain the generalization curves reported in this paper.

## 5.2 Saddle-point equations for quadratic loss

The free energy in Eq. (52) has to be evaluated at the saddle point with respect to all the RS order parameters and their Fourier conjugates. We report here the resulting equations, in the special case of quadratic loss function (10). Remark however that only the equations where $P$ appears explicitly depend on the form of the loss, and have to be modified for other choices (see Appendix G.2). The equations can be solved in steps. First, a set of $2L + 2$ nonlinear equations is used to determine the variables $\chi^{(0)}, \ldots, \chi^{(L)}$ and $\hat{\chi}^{(0)}, \ldots, \hat{\chi}^{(L)}$:

$$
\begin{aligned}
\hat{\chi}^{(0)} &= \frac{P}{N} \frac{\mu_\perp^2}{1 + \chi^\star}\,, & \chi^{(0)} &= \frac{1 - \sum_{\ell=1}^{L}[1 - \gamma_\ell g_\ell(-\gamma_\ell)]}{\hat{\chi}^{(0)} + \zeta}\,, \\
\hat{\chi}^{(\ell)} &= \frac{P}{\binom{D}{\ell}} \frac{\mu_\ell^2}{\ell!} \frac{1}{1 + \chi^\star}\,, & \chi^{(\ell)} &= \frac{N}{\binom{D}{\ell}} \frac{1 - \gamma_\ell g_\ell(-\gamma_\ell)}{\hat{\chi}^{(\ell)}}\,.
\end{aligned}
\tag{56}
$$

From the solution of Eq. (56), we can fully determine $m^{(\ell)}, \hat{m}^{(\ell)}$ according to

$$m^{(0)} = \frac{\langle y \rangle}{\mu_0}\,, \qquad m^{(\ell)} = \chi^{(\ell)} \hat{m}^{(\ell)}\,, \qquad \hat{m}^{(\ell)} = \frac{P}{\binom{D}{\ell}} \frac{\mu_\ell \tau_\ell}{\sqrt{\ell!}} \frac{\langle y\, v \rangle}{1 + \chi^\star}\,. \tag{57}$$

With all the previous values we can determine the rest of the variables through the following set of linear equations:

$$
\begin{aligned}
q^{(0)} &= \frac{\hat{q}^{(0)}}{(\zeta + \hat{\chi}^{(0)})^2} \left(1 - \sum_{\ell=1}^{L} [1 - \gamma_\ell^2 g_\ell'(-\gamma_\ell)]\right) + \sum_{\ell=1}^{L} \frac{\hat{m}^{(\ell)2} + \hat{q}^{(\ell)}}{(\zeta + \hat{\chi}^{(0)}) \hat{\chi}^{(\ell)}} \left[\gamma_\ell g_\ell(-\gamma_\ell) - \gamma_\ell^2 g_\ell'(-\gamma_\ell)\right]\,, \\
q^{(\ell)} &= \frac{N}{\binom{D}{\ell}} \frac{\hat{q}^{(0)}}{(\zeta + \hat{\chi}^{(0)}) \hat{\chi}^{(\ell)}} \left[\gamma_\ell g_\ell(-\gamma_\ell) - \gamma_\ell^2 g_\ell'(-\gamma_\ell)\right] \\
&\quad + \frac{N}{\binom{D}{\ell}} \frac{\hat{m}^{(\ell)2} + \hat{q}^{(\ell)}}{\hat{\chi}^{(\ell)2}} \left[1 + \gamma_\ell^2 g_\ell'(-\gamma_\ell) - 2\gamma_\ell g_\ell(-\gamma_\ell)\right]\,,
\end{aligned}
\tag{58}
$$

$$\hat{q}^{(0)} = \frac{P}{N}\mu_\perp^2 \frac{\langle(\mu_0 m^{(0)} - y)^2\rangle - 2\langle y\, v\rangle m^\star + q^\star}{(1 + \chi^\star)^2},$$

$$\hat{q}^{(\ell)} = \frac{P}{\binom{D}{\ell}}\frac{\mu_\ell^2}{\ell!} \frac{\langle(\mu_0 m^{(0)} - y)^2\rangle - 2\langle y\, v\rangle m^\star + q^\star}{(1 + \chi^\star)^2}.$$

Notice that, because of the conventional scalings we chose for the hat variables starting from Eq. (39) and for the definition of $\gamma_\ell$, these equations give $O(1)$ results for the order parameters $m, \chi, q$.

By numerically integrating Eq. (56), (57), (58), we obtain the theoretical curves for the generalization error in Eq. (16) and for the order parameters we report in this paper. We compare the result with numerical simulations: despite its asymptotic nature and the hypothesis of row space orthogonality, our theory works reasonably well even if $D$ is not large. The results are shown in Fig. 1, 2 ($D = 30$ in this case), where the generalization error is quantitatively predicted by the theory both when varying $P$ and $N$.

# 6 Strongly separated regimes

Our analysis relies on a number of assumptions:

1. the Gaussian ansatz on the distribution of $(v, \{\lambda^a\}_a)$ at given $F$, $\theta$ and $\mathbf{w}^a$ in the replicated partition function (39);

2. the truncation of the kernel $\mathcal{K}$ at order $L$, based on a concentration property of the matrices $C^{\odot \ell}$;

3. the fact that the row spaces of the matrices $C^{\odot \ell}$ and $C^{\odot k}$ are orthogonal for $\ell \neq k$, in order to factorize their contribution to the partition function;

4. the possibility of taking $C^{\odot \ell}$ as matrices with a spectrum asymptotically described by the Marchenko-Pastur distribution with aspect ratio $N/(D^\ell/\ell!)$;

5. the Replica Symmetric ansatz for the overlap matrices describing the teacher-student distribution;

6. the possibility of taking the saddle point on the replica parameters for large $N$, considering only the leading order in $N$, $P$ before fixing their relative scaling with $D$.

Some of these assumptions have been already discussed in the previous sections. In the following, we revise and motivate the assumptions on the matrices $C^{\odot \ell}$, namely 2-4, that can be justified if $P, N, D \to \infty$ (see Appendix D.2 for more details). Depending on the relation between the three parameters one is led to consider the following different asymptotic regimes:

(i) $N, P, D \to \infty$, $P/N \to 0$, $P/D^K$ finite; (this includes the case $N \sim D^L$ with $L > K$);

(ii) $N, P, D \to \infty$, $N/D^L$ finite, $P/N$ finite;

(iii) $N, P, D \to \infty$, $P/N \to \infty$, $N/D^L$ finite; (this includes the case $P \sim D^K$ with $K > L$).

In order to understand these regimes, we need to evaluate terms of the kind

$$k_\ell = \text{Tr} \log(a\mathbb{1} + b C^{\odot \ell}), \qquad C_{ij}^{\odot \ell} = \left(\frac{1}{D}\sum_\alpha F_{i\alpha}F_{j\alpha}\right)^\ell, \tag{59}$$

in three situations (a) $D^\ell \gg N$; (b) $D^\ell \ll N$; (c) $D^\ell \sim N$. Notice that in all cases, while the diagonal elements are $C_{ii}^{\odot\ell} = 1 + \ell O(\sqrt{1/D})$, the off-diagonal elements $C_{i \neq j}^{\odot\ell}$ are of the order $D^{-\ell/2}$. In case (a), $D^\ell \gg N$, apart for a negligible number of possible eigenvalue of order $N/D^{\ell/2}$, all the other eigenvalues are $\lambda = 1 + O(\sqrt{N/D^\ell})$, and to the leading order we simply have $k_\ell = N \log(a + b)$. If we are in the opposite situation, (b), $D^\ell \ll N$, we have only $O(D^\ell)$ non-zero eigenvalues, roughly equal to $\ell! N/D^\ell + O(\sqrt{N/D^\ell})$, and to the leading order $k_\ell = N \log(a)$. The interesting case is (c) $N = O(D^\ell)$: we have here $D^\ell$ eigenvalues of order 1 that contribute to $k_\ell$. The leading contribution can be understood writing

$$C_{ij}^{\odot\ell} = \frac{\ell!}{D^\ell} \sum_{\boldsymbol{\alpha}} F_{i,\boldsymbol{\alpha}}^{\otimes\ell} F_{j,\boldsymbol{\alpha}}^{\otimes\ell} + \text{terms with less different } \alpha\text{'s}, \tag{60}$$

where the sum includes the terms where the $\alpha's$ in the multi-index $\boldsymbol{\alpha}$ are ordered, coherently with our definition in Table 1. This leading term is a matrix of rank $\min\{N, D^\ell/\ell!\}$: the $D^\ell/\ell!$ vectors $\mathbf{F}_{\boldsymbol{\alpha}}^{\otimes\ell}$ are approximately orthogonal in $\mathbb{R}^N$, as

$$\mathbf{F}_{\boldsymbol{\alpha}}^{\otimes\ell} \cdot \mathbf{F}_{\boldsymbol{\beta}}^{\otimes\ell} = \sum_{i=1}^{N} F_{i\alpha_1} F_{i\beta_1} \cdots F_{i\alpha_\ell} F_{i\beta_\ell} = N \delta_{\alpha_1 \beta_1} \cdots \delta_{\alpha_\ell \beta_\ell} + O(N^{1/2}), \tag{61}$$

when $\boldsymbol{\alpha}$ and $\boldsymbol{\beta}$ are ordered, by law of large numbers, so that the sum of outer products $\sum_{\boldsymbol{\alpha}} \mathbf{F}_{\boldsymbol{\alpha}}^{\otimes\ell}(\mathbf{F}_{\boldsymbol{\alpha}}^{\otimes\ell})^\top$ has rank $D^\ell/\ell!$ as long as $N > D^\ell/\ell!$; if $N < D^\ell/\ell!$, this $N \times N$ matrix is full rank. Other terms with smaller number of indices in the sum lead to matrices of lower rank $r$ (with $r/N \to 0$). Moreover, due to the randomness of the $F$, the row spaces of these term are effectively orthogonal to the leading one. To understand this, take for example the case $\ell = 2$ and $N = O(D^2)$: the leading order term of the matrix $C^{\odot 2}$ has eigenvectors approximately equal to the vectors $(F_{i\beta_1} F_{i\beta_2})_i$ for $\beta_1 < \beta_2$, as

$$\sum_j \left( \frac{2}{D^2} \sum_{\alpha_1 < \alpha_2} F_{i\alpha_1} F_{i\alpha_2} F_{j\alpha_1} F_{j\alpha_2} \right) F_{j\beta_1} F_{j\beta_2} = \frac{2N}{D^2} F_{i\beta_1} F_{i\beta_2} + O(N^{1/2}/D^2). \tag{62}$$

When we apply to this vector the next-to-leading order term of the matrix $C^{\odot 2}$ we find

$$\sum_j \left( \frac{1}{D^2} \sum_{\alpha} F_{i\alpha}^2 F_{j\alpha}^2 \right) F_{j\beta_1} F_{j\beta_2} = O(N^{1/2}/D^2), \tag{63}$$

because the indices $\beta_1$ and $\beta_2$ are different and one among them remains unpaired. In this way we can say that the vectors $(F_{i\beta_1} F_{i\beta_2})_i$ are in the null space of the terms we are discarding in (60). With similar arguments, one can show that the leading terms of $C^{\odot\ell}$ and $C^{\odot k}$ have approximately orthogonal row spaces when $k \neq \ell$ and the scaling of $N$ with $D$ is fixed. We conclude that we can compute $k_\ell$ as if $C^{\odot\ell}$ were a Wishart matrix with aspect ratio $\eta_\ell = N/\binom{D}{\ell}$. The explicit formula is given in eq. (D.12), and both limits $\eta_\ell \to 0$ and $\eta_\ell \to \infty$ agree with the previous analysis of cases (a) and (b) respectively. We show in Appendix D.2 that approximating $C^{\odot\ell}$ as a Wishart matrix gives good results also for moderately large values of $N$ and $D$.

In all our three cases, most of the order parameters go to trivial limits, while only the ones corresponding to the selected scaling regime converge to non-trivial values. We report the corresponding equations in Appendix H. In this way, we are able to plot the dashed lines in Fig. 1 and 3.

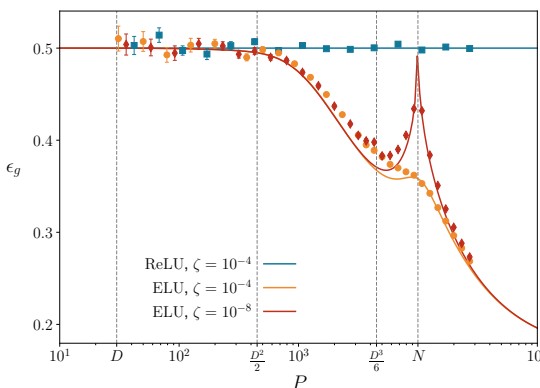

Figure 4: Generalization error vs $P$ ($D = 30$, $N = 10^4$) on classification for a *purely cubic teacher* ($\tau_3 = 1$); in blue, polynomial theory and numerical experiments for ReLU activation function (7): in this case, $\mu_3 = 0$ and the model cannot learn the cubic features, so the error remains $1/2$; in yellow and red (respectively, for $\zeta = 10^{-4}, 10^{-8}$), the case of ELU (8), for which $\mu_3 \neq 0$ and the model can learn the cubic features.

# 7 Effective theory for finite-size random features networks

In the last sections we devised a theory able to capture the relevant phenomenology of generalization in RFMs at finite values of input dimension, hidden layer width and size of the training set. Indeed, even though the asymptotic approximation leading to the system of saddle-point equations (56), (57), (58) is justified only for $N$ large and $N/D^L$ finite, the curves obtained by fixing the values of $N$, $P$ and $D$ at finite values are in accordance with numerical simulations over several orders of magnitudes of the control parameters. This occurs thanks to the fact that we kept into account quantities that scale differently with $D$, as $N/\binom{D}{\ell}$ or $P/\binom{D}{\ell}$, that are formally zero or infinity in the asymptotic regimes presented in Sec. 6.

By developing a theory from Eq. (27), we show that the RFM is in essence equivalent to a polynomial model: the student tries to tune its weights through the combinations $\mathbf{s}^{(\ell)}$ defined in (28) to fit the corresponding coefficients $\boldsymbol{\theta}^{(\ell)}$ of the teacher. This interpretation is also confirmed in the numerical experiments: see Fig. 1 (right) for the behavior of the teacher-student overlaps $m^{(\ell)}$ in the case of a quadratic teacher.

However, a crucial difference from a purely polynomial setting arises: the degree of the equivalent polynomial model is controlled by the scaling $L$ of the random features, and higher order terms in the expansion of the kernel $\mathcal{K}$ on the Hermite basis act as noise, given by Eq. (31). This eventually produces the interpolation peak in the generalization error at $N \sim P$, which would not be present for a vanilla polynomial student (see Fig. 1 and 3): in this regime, the model is using the effective noise to overfit the teacher. In terms of the order parameters, overlaps of different orders are coupled by an additional set of parameters $\chi^{(0)}$, $q^{(0)}$, related to the noise term in the equivalent polynomial model.

In summary, the learning of features of a certain order is possible as long as the number of parameters $N$ is enough: the scaling $L \sim \log N / \log D$ controls the learning process through the truncation of the kernel (25). At the same time, $P$ also plays an important role: if $K \sim \log P / \log D$ is smaller than $L$, the model only learns as a $K$-degree polynomial; on the other hand, if $K > L$, the model learns as a $L$-degree polynomial.

By choosing a polynomial teacher of arbitrary degree $B$, we are able to explore to some extent the interplay between the complexity of the data and the one of the neural network. In the case where the teacher is less complex than the network, we can see that overfitting

can occur and that overparametrization is not always optimal. This can be seen in Fig. 3. In the case of a linear teacher, if the amount of data $P$ is $O(D)$, an overparametrized network generalizes better. However, as soon as $P$ hits the quadratic regime, but is still far from enabling the network to realize that there is no quadratic feature, then overparametrization leads to overfitting and therefore the optimal $N$ is less than $P$.

Interestingly, in order for the model to learn features of order $\ell$, the activation function $\sigma$ must have a non-zero Hermite coefficient $\mu_\ell$ in Eq. (21). This can be seen from our theory by the fact that in the total teacher-student overlap $m^\star$ in Eq. (55) the single entry $m^{(\ell)}$ is weighted by the corresponding coefficient. This theoretical prediction was tested by using a cubic teacher and two different students, one with ReLU activation function and the other one with ELU: the ReLU one, which has no third order term in the Hermite basis ($\mu_3 = 0$) could not learn the teacher, while the ELU one, that does have a nonzero component ($\mu_3 \neq 0$), was able to (see Fig. 4).

# 8 Conclusions and perspectives

The approach we have explored so far provides a way to analytically evaluate the generalization performance of a RFM in the limit of large input dimension $D$, in the scaling regimes $N \sim D^L$, $P \sim D^K$.

We considered a teacher-student setting, where a shallow random features student is required to fit a polynomial teacher. The student network learns as an equivalent polynomial model with effective noise. We showed this property by expanding the kernel in feature space on a convenient basis (21).

The resulting theory is effective, in the sense that it is formulated in terms of a few collective order parameters (the teacher-student overlaps $m^{(\ell)}$, the student-student overlaps $q^{(\ell)}$, $\chi^{(\ell)}$) with a clear physical interpretation and whose values are fixed via a variational principle, as explained in Sec. 5. To perform the calculation we neglect the correlations between the student's coefficients, assuming orthogonality between the row spaces of the components $C^{\odot\ell}$ of the kernel.

We find quantitative agreement with numerical simulations, except close to the interpolation peak at $N \sim P$ in some cases (see Fig. 3, left, where this effect is more apparent). Nevertheless, even then the effective theory gives a good qualitative picture, predicting the location and the shape of the peak. See also Fig. 1, right, depicting how the teacher-student overlaps of already learned features become noisy in the interpolation regime. A precise finite-size analysis of this effect, to address the gap between theory and numerics in this regime, is left for future work.

One possible direction to continue this work is to consider how close is the learning of a fully-trained network to this model. The role of the variables $\mathbf{s}^{(\ell)}$ could play a similar role even if the values for $F_{i\alpha}$ are also learned, at least close to the lazy regime. However, what is the fate of row space orthogonality of the kernel components, which is ultimately responsible for the staircase behavior of the generalization error, for networks that are trained end-to-end in a feature learning regime?

Moreover, it would be interesting to extend our analysis to deeper models [10,74] in different scaling regimes of the dimensions. Even if the RFM, whatever the activation function of the last layer, is essentially bounded by a polynomial model, the precise shape of the kernel in cases where a deeper architecture is involved could help understanding to some extent the feature learning regimes of realistic models, in view of the discussion above. Our approach can also be extended beyond the case of unstructured input data, following for example [36,43–50,75] and, in particular, [76–78]: in those cases, we expect the intrinsic dimension of the data to

play a role similar to the parameter $D$ used here, possibly determining the order of features that a RFM can learn at given $N$ and $P$.

Finally, we mention how the replica approach we adopted here can be applied to non-convex optimization problems, at the cost of choosing a more complicated ansatz for the overlap matrices, accounting for replica symmetry breaking. Even in those cases, the replica symmetric treatment we provided can be applied as a qualitative approximation, often quantitatively correct in the teacher-student setting (that is, whenever a low-energy configuration of $\mathbf{w}$ is planted by a teacher in the energy landscape defined by the loss (9), effectively convexifying even an *a priori* non-convex problem, *i.e.* setting the problem in a replica symmetric region of a generically non-convex phase diagram – see, for example, [79, 80], studying the perceptron with hinge loss in the random labels vs. teacher-student settings).

# Acknowledgments

The authors would like to thank Pietro Rotondo, Rosalba Pacelli, Bruno Loureiro, Valentina Ros, the QBio group at ENS for discussions and suggestions. MP and FAL are grateful to the organizers and speakers of the Statistical Physics of Deep Learning summer school held in June 2022 in Como, where the idea was in part conceived.

**Funding information** The authors have been supported by a grant from the Simons Foundation (grant No. 454941, S. Franz), thanks to which most of this work was performed at LPTMS (CNRS, Université Paris-Saclay). FAL conducted part of this research within the Econophysics & Complex Systems Research Chair, under the aegis of the Fondation du Risque, the Fondation de l'École polytechnique, the École polytechnique and Capital Fund Management.

# A  Kernel on the Hermite basis

In this section we report the steps needed to obtain the expression of the feature-feature kernel in Sec. 4. The kernel to evaluate is defined as

$$
\begin{aligned}
\mathcal{K}_{ii} &= \mathbb{E}_{h_i}[\sigma(h_i)^2] = \int \frac{\mathrm{d}u}{\sqrt{2\pi C_{ii}}} e^{-\frac{u^2}{2C_{ii}}} \sigma(u)^2\,, \\
\mathcal{K}_{ij} &= \mathbb{E}_{h_i,h_j}[\sigma(h_i)\sigma(h_j)] = \int \frac{\mathrm{d}u\,\mathrm{d}v}{2\pi\sqrt{\det \bar{C}}} e^{-\frac{1}{2}(u,v)\bar{C}^{-1}(u,v)^\top} \sigma(u)\sigma(v)\,, \qquad i \neq j\,,
\end{aligned}
\tag{A.1}
$$

where

$$
\bar{C} = \begin{pmatrix} C_{ii} & C_{ij} \\ C_{ij} & C_{jj} \end{pmatrix}\,.
\tag{A.2}
$$

Using the fact that $C_{ii} \simeq C_{jj} \simeq 1$, this kernel can be written as a series of separable kernels exploiting Mehler's formula [69, 70], that we report here for convenience:

$$
\frac{1}{2\pi\sqrt{1-c^2}} e^{-\frac{1}{2}(u,v)\left(\begin{smallmatrix} 1 & c \\ c & 1 \end{smallmatrix}\right)^{-1}(u,v)^\top} = \frac{e^{-\frac{u^2}{2}}}{\sqrt{2\pi}}\frac{e^{-\frac{v^2}{2}}}{\sqrt{2\pi}} \sum_{\ell=0}^{\infty} \frac{c^\ell}{\ell!} \mathrm{He}_\ell(u)\mathrm{He}_\ell(v)\,,
\tag{A.3}
$$

from which we find Eq. (23) using the fact that, by orthogonality of the Hermite polynomials,

$$
\mathcal{K}_{ii} = \sum_{\ell=0}^{\infty} \frac{\mu_\ell^2}{\ell!}\,.
\tag{A.4}
$$

Mehler's formula, which dates back to 1866, can be viewed as an example of Mercer's decomposition [15].

# B  Hermite polynomials and Wick products

For completeness, we show in this section that, asymptotically for $D$ large,

$$\text{He}_\ell(h_i) \simeq \sum_{\alpha_1, \cdots, \alpha_\ell} \frac{F_{i\alpha_1} \cdots F_{i\alpha_\ell}}{\sqrt{D^\ell}} : x_{\alpha_1} \cdots x_{\alpha_\ell} :, \tag{B.1}$$

for $\ell \geq 1$. The equivalence follows from the generating function of the Hermite polynomials,

$$\text{He}_\ell(h_i) = \frac{d^\ell}{dt^\ell} \exp\left(t h_i - t^2/2\right)\Big|_{t=0}, \tag{B.2}$$

with $h_i = \sum_\alpha F_{i\alpha} x_\alpha / \sqrt{D}$. Defining

$$\lambda_\alpha = t \frac{F_{i\alpha}}{\sqrt{D}}, \tag{B.3}$$

we have, for $D$ large,

$$\sum_\alpha \lambda_\alpha^2 \approx t^2, \qquad \sum_\alpha \frac{F_{i\alpha} \lambda_\alpha}{\sqrt{D}} \approx t, \qquad \sum_\alpha \frac{F_{i\alpha}}{\sqrt{D}} \frac{\partial}{\partial \lambda_\alpha} \approx \frac{d}{dt}, \tag{B.4}$$

where we used repeatedly $\sum_\alpha (F_{i\alpha})^2/D \simeq 1$. The thesis follows from comparison with Eq. (29). Notice that, in the simpler case of a single standard Gaussian variable $x$, the identity $\text{He}_\ell(x) = : x^\ell :$ is exact and trivially follows from the definition of the Wick power.

# C  Evaluation of the moments of $\nu, \lambda^a$

We assume that the variables $(\nu, \{\lambda^a\})$ are normally distributed with mean and covariance

$$\mathbb{E}_\mathbf{x}[(\nu, \{\lambda^a\})] = (0, \{t^a\}), \qquad \text{cov}_\mathbf{x}[(\nu, \{\lambda^a\})] = \begin{pmatrix} \rho & M^\top \\ M & Q \end{pmatrix}, \tag{C.1}$$

where

$$
\begin{aligned}
t_a &= \mathbb{E}_\mathbf{x}[\lambda^a] = \sum_{i=1}^N \frac{w_i^a}{\sqrt{N}} \mathbb{E}_{h_i}[\sigma(h_i)], \\
\rho &= \mathbb{E}_\mathbf{x}[\nu^2] - \mathbb{E}_\mathbf{x}[\nu]^2 = \sum_{\ell=1}^B \tau_\ell^2 \frac{\|\theta^{(\ell)}\|^2}{\binom{D}{\ell}}, \\
M_a &= \mathbb{E}_\mathbf{x}[\nu \lambda^a] = \sum_{i,\ell}^{N,B} \frac{w_i^a \tau_\ell}{\sqrt{N \binom{D}{\ell}}} \sum_{\alpha_1 < \cdots < \alpha_\ell} \theta^{(\ell)}_{\alpha_1 \cdots \alpha_\ell} \mathbb{E}_\mathbf{x}[x_{\alpha_1} \cdots x_{\alpha_\ell} \sigma(h_i)], \\
Q_{ab} &= \mathbb{E}_\mathbf{x}[\lambda^a \lambda^b] - t^a t^b = \sum_{i,j=1}^N \frac{w_i^a w_j^b}{N} \mathbb{E}_{h_i, h_j}[\sigma(h_i) \sigma(h_j)] - t^a t^b.
\end{aligned}
\tag{C.2}
$$

To proceed, we make the following steps, starting from the expansion of the activation function on the Hermite basis, Eq. (21). For $t_a$ we simply observe that $\mathbb{E}_{h_i}[\sigma(h_i)] = \mu_0$. For $\rho$ we use the fact that $\mathbf{x}$ is distributed as a standard normal random vector. To deal with $Q_{ab}$ we introduce the truncation of (25). Finally, for $M_a$ we write explicitly

$$\sum_{\alpha_1 < \cdots < \alpha_k} \theta^{(k)}_{\alpha_1 \cdots \alpha_k} \mathbb{E}_\mathbf{x}[x_{\alpha_1} \cdots x_{\alpha_k} \sigma(h_i)] = \sum_{\alpha_1 < \cdots < \alpha_k} \theta^{(k)}_{\alpha_1 \cdots \alpha_k} \sum_{\ell=0}^\infty \frac{\mu_\ell}{\ell!} \mathbb{E}_\mathbf{x}[x_{\alpha_1} \cdots x_{\alpha_k} \text{He}_\ell(h_i)], \tag{C.3}$$

and we perform Wick's contractions in order to evaluate the expected value, exploiting the mapping to Wick's product explained in Appednix B. As the indices $\alpha$ of the teacher are strictly ordered, they must be paired only with the ones in the Wick product, leaving only the term $\ell = k$ in the sum over $\ell$. The number of possible contractions is $k!$, so the result is

$$t_a = \frac{\mu_0}{\sqrt{N}} \sum_{i=1}^{N} w_i^a,$$

$$M_a = \sum_i^N \frac{w_i^a}{\sqrt{N}} \sum_{\ell=1}^{B} \frac{\tau_\ell}{\binom{D}{\ell}\sqrt{\ell!}} \sum_{\boldsymbol{\alpha}} \theta_{\boldsymbol{\alpha}}^{(\ell)} F_{i,\boldsymbol{\alpha}}^{\otimes \ell}, \tag{C.4}$$

$$Q_{ab} = \frac{1}{N} \sum_{i,j=1}^{N} w_i^a w_j^b \left( \delta_{ij} \mu_{\perp,L}^2 + \sum_{\ell=1}^{L} \frac{\mu_\ell^2}{\ell!} (C_{ij})^\ell \right),$$

from which Eq. (36) follows.

# D   Results on random matrix theory

## D.1   Marchenko-Pastur distribution and Stieltjes transformation

In this section, we remind some textbook results in Random Matrix Theory we used in the main text, for the reader's convenience. First of all, random matrices of the form

$$C = FF^\top / D, \tag{D.1}$$

where $F$ is a $N \times D$ random matrix with i.i.d. entries $F_{i\alpha}$ such that $\mathbb{E}[F_{i\alpha}] = 0$, $\mathbb{E}[(F_{i\alpha})^2] = \sigma^2$, define the Wishart (or Wishart-Laguerre) ensemble. For large $N$ and $D$, parameter $\eta \equiv N/D$ finite, their spectral density follows the Marchenko-Pastur (MP) distribution,

$$\rho_{\text{MP}}(\lambda) = \begin{cases} (1 - 1/\eta) \, \delta(\lambda) + \rho_{\text{bulk}}(\lambda/\sigma^2)/\sigma^2, & \text{if } \eta > 1, \\ \rho_{\text{bulk}}(\lambda/\sigma^2)/\sigma^2, & \text{if } \eta \le 1, \end{cases} \tag{D.2}$$

with

$$\rho_{\text{bulk}}(\lambda) = \frac{\sqrt{(\lambda_+ - \lambda)(\lambda - \lambda_-)}}{2\pi\eta\lambda}, \qquad \lambda_\pm = (1 \pm \sqrt{\eta})^2, \tag{D.3}$$

with support in $\lambda_- \le \lambda \le \lambda_+$.

The MP distribution can be obtained with standard methods [81,82]. The determinant of the resolvent can be evaluated as follows:

$$\mathbb{E}\left[ \det\left( \gamma \mathbb{1}_N + \frac{FF^\top}{D} \right) \right]^{-\frac{1}{2}} = \mathbb{E} \int \frac{d\mathbf{x}}{(2\pi)^{\frac{N}{2}}} e^{-\frac{1}{2}\mathbf{x}^\top (\gamma \mathbb{1}_N + \frac{FF^\top}{D})\mathbf{x}}. \tag{D.4}$$

By Gaussian linearization,

$$\mathbb{E} \int \frac{d\mathbf{y}}{(2\pi)^{\frac{D}{2}}} \frac{d\mathbf{x}}{(2\pi)^{\frac{N}{2}}} e^{-\frac{\|\mathbf{y}\|^2}{2} - \frac{\gamma}{2}\|\mathbf{x}\|^2 + i\mathbf{x}^\top \frac{F}{\sqrt{D}}\mathbf{y}}. \tag{D.5}$$

The average over $F$ gives

$$\int \frac{d\mathbf{y}}{(2\pi)^{\frac{D}{2}}} \frac{d\mathbf{x}}{(2\pi)^{\frac{N}{2}}} e^{-\frac{\|\mathbf{y}\|^2}{2} - \frac{\gamma}{2}\|\mathbf{x}\|^2 - \frac{1}{2D}\|\mathbf{x}\|^2\|\mathbf{y}\|^2}. \tag{D.6}$$

Integrating over $\mathbf{y}$,

$$\int \frac{d\mathbf{x}}{(2\pi)^{\frac{N}{2}}} e^{-\frac{\gamma}{2}\|\mathbf{x}\|^2 - \frac{D}{2}\log(1+\|\mathbf{x}\|^2/D)}. \tag{D.7}$$

Inserting $r = \|\mathbf{x}\|^2/N$ with a Dirac delta, we can integrate over $\mathbf{x}$:

$$\int \frac{dr\,d\hat{r}}{4\pi} e^{\frac{iN\hat{r}r}{2} - \frac{N}{2}\log(i\hat{r}) - \frac{N}{2}\gamma r - \frac{N}{2\eta}\log(1+\eta r)}. \tag{D.8}$$

The integral over the Fourier variable $\hat{r}$ can be solved via asymptotic integration, the saddle-point being in $\hat{r} = -ir^{-1}$:

$$\int dr\, e^{\frac{N}{2}\left[1+\log(r) - \gamma r - \frac{1}{\eta}\log(1+\eta r)\right]}. \tag{D.9}$$

The saddle point equation in $r$ gives

$$\frac{1}{r} - \gamma - \frac{1}{1+\eta r} = 0\,, \tag{D.10}$$

with solutions

$$r_\pm = \frac{\eta - \gamma - 1 \pm \sqrt{(\eta - \gamma - 1)^2 + 4\eta\gamma}}{2\eta\gamma}\,. \tag{D.11}$$

The correct branch can be proven to be $r = r_+$. From this analysis, the relation

$$\frac{1}{N}\mathbb{E}\,\mathrm{Tr}\log(\gamma\mathbb{1} + C) = -(1-\gamma r) - \log(r) + \frac{1}{\eta}\log(1+\eta r) \tag{D.12}$$

follows. Deriving with respect to $\gamma$,

$$\frac{1}{N}\mathbb{E}\,\mathrm{Tr}(\gamma\mathbb{1} + C)^{-1} = r(\gamma)\,. \tag{D.13}$$

By definition of Stieltjes transformaiton, $r(\gamma) = g(-\gamma)$, which gives Eq. (51).

## D.2 Spectral density of $C^{\odot\ell}$

In this Appendix we discuss the spectral density of the matrices $C^{\odot\ell}$, to clarify the kind of approximation we used in the main text. We are interested to the large $N$ computation of the following traces:

$$a_\ell = \frac{1}{N}\mathrm{Tr}(\gamma_\ell\mathbb{1} + C^{\odot\ell})^{-1}\,, \qquad b_\ell = \frac{1}{N}\mathrm{Tr}\,C^{\odot\ell}(\gamma_\ell\mathbb{1} + C^{\odot\ell})^{-1}\,, \tag{D.14}$$

under the hypothesis that $\eta_L = N/\binom{D}{L}$ remain finite. We anticipate that $\gamma_\ell$ given by (48) either remain finite (if $P/N$ remains finite) or tends to infinity (if $P/N \to \infty$) in that limit. As we have already discussed, for $\ell > L$, the matrix $C^{\odot\ell}$ is fully ranked, with diagonal elements close to one and off-diagonal elements of order $D^{-\ell/2}$: all eigenvalues will be equal to one up to a negligible correction. For that reason we could neglect off-diagonal terms for $\ell > L$ and $a_\ell \approx b_\ell \approx (1+\gamma_\ell)^{-1}$. For $\ell < L$ conversely, the matrix has rank $D^\ell$ at most, and it is easy to see that its max eigenvalue cannot be larger that $N\max_i\left(\frac{1}{D}\sum_\alpha F_{i,\alpha}^2\right)^\ell = N(1+O(\sqrt{\log(N)/D}))$.[4] We get therefore

$$\frac{1}{N}\left((N-D^\ell)/\gamma_\ell + D^\ell/(\gamma_\ell + N)\right) \le a_\ell \le \frac{1}{\gamma_\ell}\,, \qquad 0 \le b_\ell \le \frac{D^\ell}{N}\frac{N}{\gamma_\ell + N}\,. \tag{D.15}$$

---

[4] $\lambda_{max} = \max_{v|v^2=1}\frac{1}{D^\ell}\sum_{\alpha_1,\dots,\alpha_\ell}\left(\sum_i v_i F_{i,\alpha_1}\dots F_{i,\alpha_\ell}\right)^2 \le N\sum_i v_i^2\left(\frac{1}{D}\sum_\alpha F_{i,\alpha}^2\right)^\ell$.

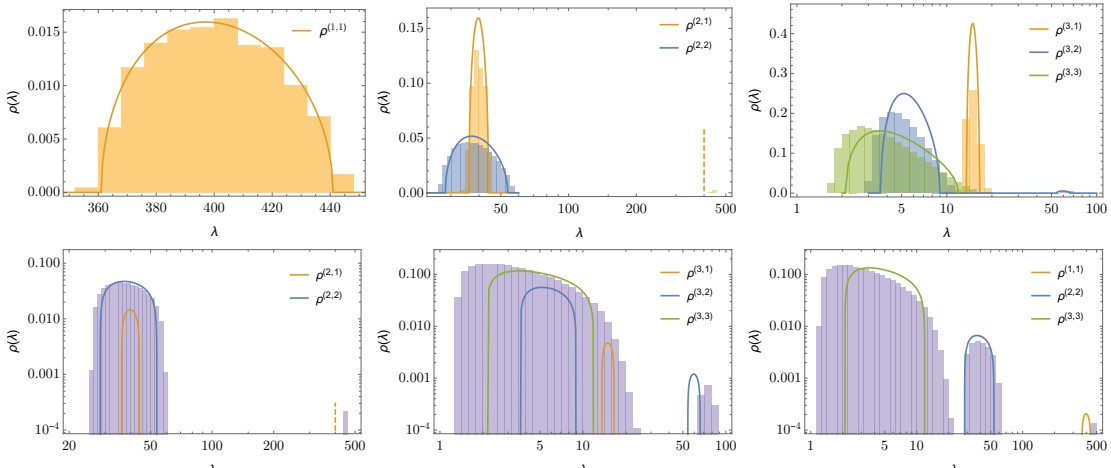

Figure 5: **Top row** – empirical (30 instances, $D = 20$, $N = D^3$) vs. analytical (MP) distributions of the non-zero eigenvalues of the matrices defined in Sec. D.2: $C^{(1,1)}$ (left), $C^{(2,1)}/D$, $C^{(2,2)}$ (center), $C^{(3,1)}/D^2$, $3C^{(3,2)}/D$, $C^{(3,3)}$ (right). **Bottom row** – comparison of the analytical curves with the empirical distribution (notice the log scale on the axes) of $C^{\odot 2}$ (left), $C^{\odot 3}$ (center) and $C^{\odot 1} + C^{\odot 2} + C^{\odot 3}$ (right); analytical curves in the bottom row are rescaled in such a way that the sum of the densities in each panel is normalized.

It remains to be discussed the only non trivial case: $\ell = L$ In that case, we can decompose the matrix $C^{\odot L}$ as a matrix with rank $\min\{N, \binom{D}{L}\}$ and spectrum asymptotically distributed accordin to the Marchenko-Pastur law with parameter $\eta_L$, plus a contribution with rank at most $D^{L-1}$ which for reasoning similar to the previous case, do not contribute to $a_L$ and $b_L$ in the thermodynamic limit.

We would like now to show, that even for moderate values of $N$ and $D$, neglecting all the subleading contributions provides an excellent approximation to the spectrum. To fix ideas, let us consider $L = 3$ ($N \sim D^3$), so that we consider the matrices

$$C^{\odot 1} = C^{(1,1)}, \qquad C^{\odot 2} = \frac{1}{D} C^{(2,1)} + C^{(2,2)}, \qquad C^{\odot 3} = \frac{1}{D^2} C^{(3,1)} + \frac{3}{D} C^{(3,2)} + C^{(3,3)}, \quad \text{(D.16)}$$

where (we use the label $(\ell, k)$, where $\ell$ is the corresponding exponent in $C^{\odot \ell}$, and $k$ the number of different summation indices)

$$
\begin{aligned}
C_{ij}^{(1,1)} &= \frac{1}{D} \sum_\alpha F_{i\alpha} F_{j\alpha} = C_{ij}, \\
C_{ij}^{(2,1)} &= \frac{1}{D} \sum_\alpha F_{i\alpha}^2 F_{j\alpha}^2, \\
C_{ij}^{(2,2)} &= \frac{2}{D^2} \sum_{\alpha < \beta} F_{i\alpha} F_{i\beta} F_{j\alpha} F_{j\beta}, \\
C_{ij}^{(3,1)} &= \frac{1}{D} \sum_\alpha F_{i\alpha}^3 F_{j\alpha}^3, \\
C_{ij}^{(3,2)} &= \frac{1}{D^2} \sum_{\alpha \neq \beta} F_{i\alpha}^2 F_{i\beta} F_{j\alpha}^2 F_{j\beta}, \\
C_{ij}^{(3,3)} &= \frac{6}{D^3} \sum_{\alpha < \beta < \gamma} F_{i\alpha} F_{i\beta} F_{i\gamma} F_{j\alpha} F_{j\beta} F_{j\gamma}.
\end{aligned}
\qquad \text{(D.17)}
$$

We can say the following on the matrices $C^{(\ell,k)}$ when $N$, $D$ are both (generically) large:

- $C^{(1,1)} = C$ has a Marchenko-Pastur (MP) spectrum with parameter $\eta_1 = N/D$ and $\sigma^2 = 1$, with $D$ bulk eigenvalues $\lambda = N/D + O(\sqrt{\frac{N}{D}})$ (and $N - D$ zero eigenvalues).

- $C^{(2,1)}$ can be written as

$$C_{ij}^{(2,1)} \simeq 1 + \frac{1}{D}\sum_\alpha (\Delta_{i\alpha}\Delta_{j\alpha}), \tag{D.18}$$

  where $\Delta_{i\alpha} = F_{i\alpha}^2 - \mathbb{E}[F_{i\alpha}^2] = F_{i\alpha}^2 - 1$. Notice that $\mathbb{E}[\Delta_{i\alpha}^2] = 2$. From this, it follows that $C^{(2,1)}$ has an MP spectrum with parameter $\eta_1$ and $\sigma^2 = 2$, with $D$ bulk eigenvalues $O(\sigma^2\eta_1)$, plus an additional outlier eigenvalue of order $N$ (due to the finite mean); however, in $C^{\odot 2}$ this matrix is scaled by an additional factor of $1/D$, so it contributes to the sum with $D$ eigenvalues $O(2N/D^2)$ and an outlier $O(N/D)$.

- $C^{(2,2)}$ has an MP spectrum with parameter $\eta_2 = 2N/D^2$ and $\sigma^2 = 1$, with $D^2/2$ bulk eigenvalues $O(\eta_2)$.

- $C^{(3,1)}$ has an MP spectrum with parameter $\eta_1$ and $\sigma^2 = 15$, with $D$ bulk eigenvalues $O(\eta_1)$; however, in $C^{\odot 3}$ this matrix is scaled by an additional factor of $1/D^2$, so it contributes to the sum with $D$ eigenvalues $O(N/D^3)$.

- $C^{(3,2)}$ can be written as

$$C^{(3,2)} \simeq \frac{1}{D^2}\sum_{\alpha\neq\beta}\Delta_{i\alpha}F_{i\beta}\Delta_{j\alpha}F_{j\beta} + \frac{1}{D}\sum_\alpha F_{i\alpha}F_{j\alpha}. \tag{D.19}$$

  The first addendum (notice that the double sum is not symmetric) has an MP spectrum with parameter $N/D^2$ and $\sigma^2 = 2$, with $D^2$ eigenvalues $O(2N/D^2)$, while the second addendum is $C$; however, in $C^{\odot 3}$ they are both scaled by a factor $3/D$, so they contribute to the sum with $D^2$ eigenvalues $O(6N/D^3)$ and with $D$ eigenvalues $O(3N/D^2)$.

- $C^{(3,3)}$ has an MP spectrum with parameter $\eta_3 = 6N/D^3$ and $\sigma^2 = 1$, with $D^3/6$ bulk eigenvalues $O(\alpha_3)$.

This heuristics is compared with numerical results in Fig. 5, which shows a remarkable accordance. In the main text, we took the approximation $C^{\odot\ell} \simeq C^{(\ell,\ell)}$, and considered the row spaces of $C^{\odot\ell}$ for different $\ell$ as orthogonal: in Fig. 5, bottom right, we show how the spectrum of a sum of the full matrices $C^{\odot\ell}$ is reasonably approximated by the sum of the (analytical) spectra of the corresponding $C^{(\ell,\ell)}$ matrices, validating our approach.

# E Determinant of sum of matrices with orthogonal row spaces

In this section we derive Eq. (44). Let us take the $N \times N$ matrix given by

$$K = a\mathbb{1} + \sum_{\ell=1}^L b_\ell C_\ell, \tag{E.1}$$

where the matrices $C_\ell$ are such that $\mathrm{rank}(C_\ell) = r_\ell$, $\sum_\ell r_\ell \leq N$ and their row spaces $\mathcal{R}_\ell$ (that is, the orthogonal complements to their null spaces) are mutually orthogonal ($\mathcal{R}_\ell \perp \mathcal{R}_k$ for $k \neq \ell$). Then,

$$\det K = a^{N-\sum_\ell r_\ell}\prod_\ell \det_\parallel^{(\ell)}(a\mathbb{1} + b_\ell C_\ell), \tag{E.2}$$

where $\det_{\parallel}^{(\ell)}(\cdot)$ is the determinant restricted to the row space of $C_\ell$:

$$\det_{\parallel}^{(\ell)}(a\mathbb{1} + b_\ell C_\ell) = \prod_{\alpha=1}^{r_\ell}(a + b_\ell \lambda_\alpha), \tag{E.3}$$

with $\lambda_\alpha$ the non-zero eigenvalues of $C_\ell$. Eq. (E.2) can be proven by noticing that, if $\{e_\ell^\alpha\}_{\alpha=1}^{r_\ell}$ is a basis of $\mathcal{R}_\ell$ and $\{e_\perp^\alpha\}_{\alpha=1}^{N-\sum_\ell r_\ell}$ a basis of $(\bigcup_\ell \mathcal{R}_\ell)^\perp$, the set $(\bigcup_\ell \{e_\ell^\alpha\}) \bigcup \{e_\perp^\alpha\}$ is a basis of $\mathbb{R}^N$ in which the matrix $K$ is in block-diagonal form. Moreover, from Eq. (E.3)

$$\det_{\parallel}^{(\ell)}(a\mathbb{1} + b_\ell C_\ell) = \det(a\mathbb{1} + b_\ell C_\ell)a^{-(N-r_\ell)}, \tag{E.4}$$

so we can conclude that

$$\det K = a^{N(1-L)} \prod_\ell \det(a\mathbb{1} + b_\ell C_\ell). \tag{E.5}$$

# F   Traces over RS matrices

In this section we derive Eq. (47). We need to evaluate

$$\mathrm{Tr}\log\left(A \otimes \mathbb{1}_N + B \otimes C^{\odot\ell}\right), \tag{F.1}$$

where $A$, $B$ are RS $n \times n$ matrices. We can write

$$A \otimes \mathbb{1}_N + B \otimes C^{\odot\ell} = (B \otimes \mathbb{1}_N)\left(B^{-1}A \oplus C^{\odot\ell}\right), \tag{F.2}$$

where the Kronecker sum is defined as

$$B^{-1}A \oplus C^{\odot\ell} = B^{-1}A \otimes \mathbb{1}_N + \mathbb{1}_n \otimes C^{\odot\ell}. \tag{F.3}$$

The eigenvalues of a Kronecker sum are the sums of the eigenvalues of the addenda. Calling $\sigma_a$ the eigenvalues of $B^{-1}A$ and $\lambda_i$ the eigenvalues of $C^{\odot\ell}$, this means that

$$\log\det(B^{-1}A \oplus C^{\odot\ell}) = \sum_{a,i}\log(\sigma_a + \lambda_i). \tag{F.4}$$

Given that $B^{-1}A$ is RS, it has 2 different eigenvalues, $\sigma$ with multiplicity $n-1$ and $\sigma + n\tilde{\sigma}$ with multiplicity 1, so that for small $n$

$$\log\det(B^{-1}A \oplus C^{\odot\ell}) = n\sum_i \log(\sigma + \lambda_i) + n\sum_i \frac{\tilde{\sigma}}{\sigma + \lambda_i}. \tag{F.5}$$

In total we get

$$\mathrm{Tr}\log\left(A \otimes \mathbb{1}_N + B \otimes C^{\odot\ell}\right) = nN\log b + nN\frac{\tilde{b}}{b} + n\sum_i \log(\sigma + \lambda_i) + n\sum_i \frac{\tilde{\sigma}}{\sigma + \lambda_i}. \tag{F.6}$$

Using the RS algebra, we know that $\sigma = a/b$, $\tilde{\sigma} = (b\tilde{a} - a\tilde{b})/b^2$, so that

$$\mathrm{Tr}\log\left(A \otimes \mathbb{1}_N + B \otimes C^{\odot\ell}\right) = n\,\mathrm{Tr}\log(a\mathbb{1} + bC^{\odot\ell}) + n\tilde{a}\,\mathrm{Tr}(a\mathbb{1} + bC^{\odot\ell})^{-1} + n\tilde{b}\,\mathrm{Tr}[C^{\odot\ell}(a\mathbb{1} + bC^{\odot\ell})^{-1}]. \tag{F.7}$$

It only remains to find $a$, $\tilde{a}$, $b$, $\tilde{b}$:

$$a = \beta(\zeta + \hat{\chi}^{(0)}), \qquad \tilde{a} = -\beta^2 \hat{q}^{(0)}, \qquad b = \beta\hat{\chi}^{(\ell)}/\eta_\ell, \qquad \tilde{b} = -\beta^2[\hat{q}^{(\ell)} + (\hat{m}^{(\ell)})^2]/\eta_\ell. \tag{F.8}$$

We define $\gamma_\ell = a/b = \eta_\ell(\zeta + \hat{\chi}^{(0)})/\hat{\chi}^{(\ell)}$ to get Eq. (47).

# G Replica-symmetric free energy

In this section we report the main steps to obtain the terms $S_M$ and $S_P$ in Eq. (53) and (54), that is the measure and pattern contributions to the free energy.

## G.1 Measure contribution

By plugging the RS ansatz (45), (46) and Eq. (47) in Eq. (42), we readily obtain

$$
\begin{aligned}
S_M = &-n\beta \sum_{\ell=1}^{L} \frac{m^{(\ell)} \hat{m}^{(\ell)}}{\eta_\ell} + \frac{n}{2} \sum_{\ell=0}^{L} \frac{1}{\eta_\ell} [\chi^{(\ell)} \hat{\chi}^{(\ell)} + \beta(q^{(\ell)} \hat{\chi}^{(\ell)} - \chi^{(\ell)} \hat{q}^{(\ell)})] \\
&- \frac{n}{2} \log(\beta(\zeta + \hat{\chi}^{(0)})) + \frac{\beta n(1-L)}{2} \frac{\hat{q}^{(0)}}{\zeta + \hat{\chi}^{(0)}} - \frac{n}{2N} \sum_{\ell=1}^{L} \mathrm{Tr} \log(\mathbb{1} + C^{\odot \ell}/\gamma_\ell) \\
&+ \frac{\beta n}{2N} \sum_{\ell=1}^{L} \eta_\ell \frac{\hat{q}^{(0)}}{\hat{\chi}^{(\ell)}} \mathrm{Tr}(\gamma_\ell \mathbb{1} + C^{\odot \ell})^{-1} + \frac{\beta n}{2N} \sum_{\ell=1}^{L} \frac{\hat{q}^{(\ell)} + (\hat{m}^{(\ell)})^2}{\hat{\chi}^{(\ell)}} \mathrm{Tr}[C^{\odot \ell} (\gamma_\ell \mathbb{1} + C^{\odot \ell})^{-1}].
\end{aligned}
\tag{G.1}
$$

We obtain Eq. (53) by keeping the leading order terms for $\beta$ large and using Eq. (50).

## G.2 Pattern contribution

$S_P$ is a function only of the order parameters:

$$
S_P = \log\left[ \int d\nu \prod_{a=1}^{n} d\lambda^a p(\nu, \{\lambda^a\}) \int dy\, p(y|\nu) e^{-\beta \sum_a \mathcal{L}(y, \lambda^a)} \right],
$$
$$
p(\nu, \{\lambda^a\}) = \mathcal{N}\left( (\nu, \{\lambda^a\}) \,\bigg|\, (0, \{t^a\}), \begin{pmatrix} 1 & M^\top \\ M & Q \end{pmatrix} \right).
\tag{G.2}
$$

With the RS ansatz and for small $n$,

$$
\begin{aligned}
S_P = &\log\left[ \int dy\, d\nu \prod_{a=1}^{n} d\lambda^a p(y|\nu) e^{-\frac{\nu^2}{2} + \beta \frac{m^\star \nu}{\chi^\star} \sum_a \lambda^a - \frac{\beta}{2\chi^\star} \sum_a \lambda_a^2 - \beta \sum_a \mathcal{L}(y, \lambda^a + t^\star) - \beta^2 \frac{m^{\star 2} - q^\star}{2\chi^{\star 2}} \sum_{a,b} \lambda^a \lambda^b} \right] \\
&- \frac{n}{2} \log(2\pi) - \frac{1}{2} \log\det \begin{pmatrix} 1 & M^\top \\ M & Q \end{pmatrix}.
\end{aligned}
\tag{G.3}
$$

To factorize the integral over replicas we use the Hubbard-Stratonovich transformation

$$
e^{-\beta^2 \frac{m^{\star 2} - q^\star}{2\chi^{\star 2}} \sum_{a,b} \lambda^a \lambda^b} = \mathbb{E}_\xi\, e^{\beta \frac{\sqrt{q^\star - m^{\star 2}}}{\chi^\star} \sum_a \lambda^a \xi},
\tag{G.4}
$$

obtaining, to leading order in $n$,

$$
S_P = -\frac{n}{2} \log \frac{\chi^\star}{\beta} - \frac{n\beta}{2} \frac{q^\star}{\chi^\star} + n \mathbb{E}_\xi \int dy\, D\nu\, p(y|\nu) \log \int d\lambda\, e^{\beta\left(\sqrt{q^\star - m^{\star 2}} \xi + m^\star \nu\right) \frac{\lambda}{\chi^\star} - \frac{\beta \lambda^2}{2\chi^\star} - \beta \mathcal{L}(y, \lambda + t^\star)}.
\tag{G.5}
$$

For our choice of loss (10) and for $\beta$ large, we obtain Eq. (54).

For a generic choice of loss $\mathcal{L}$, the integral in $\lambda$ in Eq. (G.5) can still be evaluated asymptotically for large $\beta$. The saddle point in $\lambda$ is given by

$$
\lambda^\star = \underset{\lambda}{\mathrm{argmin}} \left[ \frac{\lambda^2}{2\chi^\star} + \mathcal{L}(y, \lambda + t^\star) - \frac{\sqrt{q^\star - m^{\star 2}} \xi + m^\star \nu}{\chi^\star} \lambda \right],
\tag{G.6}
$$

that is by the solution of the stationary equation

$$\lambda + \chi^\star \frac{\partial}{\partial \lambda} \mathcal{L}(y, \lambda + t^\star) = \sqrt{q^\star - m^{\star 2}} \xi + m^\star v. \tag{G.7}$$

For any choice of $\mathcal{L}$, this equation gives the value of $\lambda^\star$ as a function of $y$, $v$, $\xi$ and the order parameters. By substituting this value in (G.5) we obtain a generalized form of $S_P$ valid for any loss. By differentiating with respect to the order parameters, we obtain saddle point equations valid for any loss, generalizing the ones for the hat variables reported in Sec. 5.2.

## H   Asymptotic limits of the saddle-point equations

The system of saddle-point equations can be studied in different asymptotic limits, as we anticipated in Sec. 6:

(i)  $N, P, D \to \infty$, $P/N \to 0$, $P/D^K$ finite;

(ii)  $N, P, D \to \infty$, $N/D^L$ finite, $P/N$ finite;

(iii)  $N, P, D \to \infty$, $P/N \to \infty$, $N/D^L$ finite.

### H.1   Case (i)

In the limit where $N$ scales faster to infinity than $P$, Eq. (56) reduces to

$$\hat{\chi}^{(0)} \to 0, \qquad\qquad\qquad \chi^{(0)} \to \frac{1}{\zeta},$$

$$\hat{\chi}^{(\ell)} \to \begin{cases} \infty, & \text{for } \ell < K, \\ \frac{P}{\binom{D}{K}} \frac{\mu_K^2}{K!(1+\chi^\star)}, & \text{for } \ell = K, \\ 0, & \text{for } \ell > K, \end{cases} \qquad \chi^{(\ell)} \to \begin{cases} 0, & \text{for } \ell < K, \\ \frac{1}{\hat{\chi}^{(K)}+\zeta}, & \text{for } \ell = K, \\ \frac{1}{\zeta}, & \text{for } \ell > K, \end{cases} \tag{H.1}$$

where we used the asymptotic results for the Stieltjes transformation of the Marchenko-Pastur distribution,

$$1 - \gamma_\ell g(-\gamma_\ell; \eta_\ell) \sim \begin{cases} \frac{1}{\eta_\ell}, & \text{for } \ell < K, \\ \frac{1}{\eta_K + \gamma_K}, & \text{for } \ell = K, \\ \frac{1}{\gamma_\ell}, & \text{for } \ell > K. \end{cases} \tag{H.2}$$

Notice that now, consistently,

$$\chi^\star = \frac{\mu_{\perp,K}^2}{\zeta} + \frac{\mu_K^2}{K!} \chi^{(K)}, \tag{H.3}$$

because $\mu_{\perp,L}^2$ recombines with the terms coming from $K < \ell \le L$ to give $\mu_{\perp,K}^2$. Eq. (57) reduces to

$$m^{(0)} = \frac{\langle y \rangle}{\mu_0},$$

$$\hat{m}^{(\ell)} \to \begin{cases} \infty, & \text{for } \ell < K, \\ \frac{P}{\binom{D}{K}} \frac{\mu_K \tau_K}{\sqrt{K!}} \frac{\langle yv \rangle}{1+\chi^\star}, & \text{for } \ell = K, \\ 0, & \text{for } \ell > K, \end{cases} \tag{H.4}$$

$$m^{(\ell)} \to \begin{cases} \sqrt{\ell!} \frac{\tau_\ell}{\mu_\ell} \langle yv \rangle, & \text{for } \ell < K, \\ \sqrt{K!} \frac{\tau_K}{\mu_K} \langle yv \rangle (1 - \zeta \chi^{(K)}), & \text{for } \ell = K, \\ 0, & \text{for } \ell > K, \end{cases}$$

while Eq. (58) becomes

$$\hat{q}^{(0)} \to 0 \,,$$

$$\hat{q}^{(\ell)} \to \begin{cases} \infty \,, & \text{for } \ell < K \,, \\ \frac{P}{\binom{D}{K}} \frac{\mu_K^2}{K!} \frac{\langle (\mu_0 m^{(0)} - y)^2 \rangle - 2\langle y\,v \rangle m^\star + q^\star}{(1+\chi^\star)^2} \,, & \text{for } \ell = K \,, \\ 0 \,, & \text{for } \ell > K \,, \end{cases}$$

$$q^{(0)} \to 0$$

$$q^{(\ell)} \to \begin{cases} \ell! \frac{\tau_\ell^2}{\mu_\ell^2} \langle y\,v \rangle^2 \,, & \text{for } \ell < K \,, \\ \frac{(\hat{m}^{(K)2} + \hat{q}^{(K)})}{(\hat{\chi}^{(K)} + \zeta)^2} \,, & \text{for } \ell = K \,, \\ 0 \,, & \text{for } \ell > K \,, \end{cases}$$

(H.5)

where now

$$q^\star = \langle y\,v \rangle^2 \sum_{\ell=1}^{K-1} \tau_\ell^2 + \frac{\mu_K^2}{K!} q^{(K)} \,, \qquad m^\star = \langle y\,v \rangle \sum_{\ell=1}^{K-1} \tau_\ell^2 + \frac{\mu_K \tau_K}{\sqrt{K!}} m^{(K)} \,. \tag{H.6}$$

## H.2  Case (ii)

In the limit where both $P$ and $N$ scale in the the same way, $N \sim P \sim O(D^L)$, we have, for $0 < \ell < L$,

$$\hat{\chi}^{(\ell)} \to \infty \,, \qquad \hat{m}^{(\ell)} \to \infty \,, \qquad \hat{q}^{(\ell)} \to \infty \,,$$

$$\chi^{(\ell)} \to 0 \,, \qquad m^{(\ell)} \to \sqrt{\ell!} \frac{\tau_\ell}{\mu_\ell} \langle y\,v \rangle \,, \qquad q^{(\ell)} \to \ell! \frac{\tau_\ell^2}{\mu_\ell^2} \langle y\,v \rangle^2 \,. \tag{H.7}$$

For the other parameters we need to solve the equations for $\chi$

$$\hat{\chi}^{(0)} = \frac{P}{N} \frac{\mu_{\perp,L}^2}{1+\chi^\star} \,, \qquad\qquad \chi^{(0)} = \frac{\gamma_L g_L(-\gamma_L)}{\hat{\chi}^{(0)} + \zeta} \,,$$

$$\hat{\chi}^{(L)} = \frac{P}{\binom{D}{L}L!} \frac{\mu_L^2}{1+\chi^\star} \,, \qquad\qquad \chi^{(L)} = \frac{N}{\binom{D}{L}} \frac{1 - \gamma_L g_L(-\gamma_L)}{\hat{\chi}^{(L)}} \,,$$

(H.8)

for $m$,

$$m^{(0)} = \langle y \rangle / \mu_0 \,, \qquad m^{(L)} = \chi^{(L)} \hat{m}^{(L)} \,, \qquad \hat{m}^{(L)} = \frac{P}{\binom{D}{L}} \frac{\mu_L \tau_L}{\sqrt{L!}} \frac{\langle y\,v \rangle}{1+\chi^\star} \,, \tag{H.9}$$

and for $q$

$$\hat{q}^{(0)} = \frac{P}{N} \mu_{\perp,L}^2 \frac{\langle (\mu_0 m^{(0)} - y)^2 \rangle - 2\langle y\,v \rangle m^\star + q^\star}{(1+\chi^\star)^2} \,,$$

$$\hat{q}^{(L)} = \frac{P}{\binom{D}{L}} \frac{\mu_L^2}{L!} \frac{\langle (\mu_0 m^{(0)} - y)^2 \rangle - 2\langle y\,v \rangle m^\star + q^\star}{(1+\chi^\star)^2} \,,$$

$$q^{(0)} = \frac{\hat{q}^{(0)}}{(\zeta + \hat{\chi}^{(0)})^2} \gamma_L^2 g_L'(-\gamma_L) + \frac{\hat{m}^{(L)2} + \hat{q}^{(L)}}{(\zeta + \hat{\chi}^{(0)})\hat{\chi}^{(L)}} \left[ \gamma_L g_L(-\gamma_L) - \gamma_L^2 g_L'(-\gamma_L) \right] \,, \tag{H.10}$$

$$q^{(L)} = \frac{N}{\binom{D}{L}} \frac{\hat{q}^{(0)}}{(\zeta + \hat{\chi}^{(0)})\hat{\chi}^{(L)}} \left[ \gamma_L g_L(-\gamma_L) - \gamma_L^2 g_L'(-\gamma_L) \right]$$

$$+ \frac{N}{\binom{D}{L}} \frac{\hat{m}^{(L)2} + \hat{q}^{(L)}}{\hat{\chi}^{(L)2}} \left[ 1 + \gamma_L^2 g_L'(-\gamma_L) - 2\gamma_L g_L(-\gamma_L) \right] \,.$$

The values $\chi^\star$, $m^\star$ and $q^\star$ are consistent with their definition. At variance with case (i), $\chi^{(0)}$ and $q^{(0)}$ have non-trivial values, responsible for the interpolation peak appearing in this regime. Notice that their value is controlled explicitly by the regularizer $\zeta$: the lower it is, the sharper is the peak. Moreover, the spectral function relative to the active component, $g_L$, also gives a non-trivial contribution.

### H.3  Case (iii)

In the limit where $P$ is scaling faster than $N$ to infinity, we have that for all $0 < \ell < L$ the order parameters behave as in Eq. (H.7), meaning that the degree-$L$ student learns perfectly all the terms of the teacher of degree less then $L$, as the amount of training data $P$ is effectively infinite. In this case

$$\gamma_L = \frac{L!\mu_{\perp,L}^2}{\mu_L^2},\tag{H.11}$$

and we have $\chi^{(L)}$, $\hat{\chi}^{(L)} \to 0$; $\hat{q}^{(0)}$, $\hat{q}^{(L)} \to \infty$ and

$$
\begin{aligned}
m^{(L)} &= \eta_L \langle y\nu\rangle \sqrt{L!}\frac{\tau_L}{\mu_L}(1-\gamma_L g_L(-\gamma_L)),\\
q^{(0)} &= \eta_L \langle y\nu\rangle^2 \frac{\tau_L^2}{\mu_{\perp,L}^2}\left[\gamma_L g_L(-\gamma_L) - \gamma_L^2 g_L'(-\gamma_L)\right],\\
q^{(L)} &= \eta_L \langle y\nu\rangle^2 L!\frac{\tau_L^2}{\mu_L^2}\left[1+\gamma_L^2 g_L'(-\gamma_L) - 2\gamma_L g_L(-\gamma_L)\right].
\end{aligned}
\tag{H.12}
$$

## I  Numerical experiments

All numerical experiments were done in Python using JAX [83], to generate the synthetic random data, and scikit-learn [62], to optimize the parameters. The optimizer has a simple analytic form given by (18). Nevertheless, it is potentially inefficient to implement the formula naively, as it would require the inversion of a very large matrix. Since we used very large values of $N$ and $P$, we performed the ridge regression with the function sklearn.linear_model.Ridge. In this way we could explore regimes of $N, P$ up to order $D^3$.

Almost all numerical experiments were performed with $D = 30$. In most of the simulations we sampled 50 times for each combination of $N, P, D$. For the right panel of Figure 3 we used a larger number of samples since in that case both $D = 30$ and $P = 40 \sim 400$ were small, hence the generalization error had higher variability. For $N < 3000$ we used $500, 200, 300$ samples respectively for $P = 40, 200, 400$. For $N > 3000$ we used $100, 100, 50$ samples respectively for $P = 40, 200, 400$.

A GitHub repository collecting the code needed to reproduce the figures of this paper (both numerical experiments and theoretical curves from the integration of the saddle-point equations) can be found at [84].

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
