# Peer review of "Random features and polynomial rules"

_SciPost Physics, doi:SciPost Phys. 18, 039 (2025)_

## Round 1 · Referee Report · Anonymous (Referee 1) · 2024-7-22

Strengths

- The authors provide a tight asymptotic characterization of the learning of Random Features Models (RFM) on a random polynomial target function, in various data/width/dimension regimes.

-They outline and identify data (resp. width) limited regimes where the RFM reduces to a kernel (resp. polynomial regression) method, as well as a non-trivial width~data regime, exhibiting in particular an interpolation peak phenomenon.

-The derivation relies on the replica method from statistical physics and several random matrix theory arguments and approximations. All steps are rather clearly justified, motivated, and discussed.

- The analytical findings are supported by convincing numerics.

- The consequences/takeaways of the analytical results are discussed, notably in terms of overfitting and expressive power.

Weaknesses

I am listing a few minor points, typos or questions in the "changes requested" section. Here, I am listing some of my main questions and concerns.

- l. 154 (definition of the teacher). The authors consider a random teacher function, which doesn't allow to investigate simple and natural target functions such as ||x||^2, Hermite polynomials or spherical harmonics. This prevents in-depth connection and comparison to related results on kernel learning, e.g. [22]. How important is averaging over the teacher in the derivation ? [57] (which I am aware is contemporaneous to the reviewed paper, and has a arXiv released after the first arXiv version of the present work) for example seem to be able to accomodate deterministic targets, at least for the learnable polynomial space.

- In the otherwise rather complete related works section, l.122, maybe the works of [Zavatone-Veth and Pehlevan, 2024] and [Schroder et al, 2024] on deep structured RFMs, and that of [Defilippis, Loureiro, Misiakiewicz 2024] on dimension-free characterizations of RFMs could be included. I am aware some of these works are contemporaneous or appeared after the release of the first arXiv version of the present work, though prior to the present submission/version, and would leave the decision to the authors and the editor.

- p.10 : I understood the discussion, but it should ideally be clarified. In particular, could the equivalent model be written in terms of Hermite polynomials instead of Wick products, to connect with related equivalent maps e.g. [22]? Also, adding a short appendix explicitly showing how the features (26) admit population covariance (24), if this is the case, would be helpful.

- Some technical approximations (l.307, l. 243-246, further elaborated in the questions in "requested changes") are merely stated without sufficient discussion. I have not fully understood how these statements are supported, or if they are heuristic assumptions, and feel like further discussion is needed in these passages.

Report

I am overall in favor of acceptance. While I have not carefully gone through every reported technical steps, the overall derivation seems scientifically sound. The question explored is of interest, and my concerns are primarily on some aspects of the exposition of the results, although the overall quality of the writing is largely good and clear.

Requested changes

I am listing below a number of typos, comments, and minor questions.

-l.115 "as long as with finite dimensional outputs"
- l. 187 missing "of"
-l.201 Slightly awkward phrasing, maybe "since x is a test pt, and is thus uncorrelated with ...." would be simpler and clearer.
-l. 226 missing "e"
-l.243-246 Is there any (even non-rigorous) reason to expect the rank to be given by this minimum ? Isn't it in full generality just an upper bound ? More discussion would be helpful.
- Similarly, the statement that off-diagonal elements don't affect eigenvalues/vectors is a bit too fast, and further discussion would be helpful to support this.
- l.343 incomplete sentence.
-l.417 Instead of "overfitting the effective noise", isn't the model rather using the effective noise to overfit the teacher ? The two phenomena are different.
- (55) Though I understand the approximation of neglecting diagonal terms, I am not sure to understand why the l-th Hadamard power of C can be thought of as Wishart ? In particular, it seems the corresponding matrix involved in the product doesn't have Gaussian entries, and the entries are also not mutually independent ? Perhaps more discussion would help.
-l.307 Why are the row spaces assumed orthogonal ? Again, more discussion would prove useful.

Recommendation

Publish (meets expectations and criteria for this Journal)

  • validity: high
  • significance: good
  • originality: good
  • clarity: good
  • formatting: excellent
  • grammar: good

Author:  Mauro Pastore  on 2024-11-07  [id 4944]

(in reply to Report 1 on 2024-07-22)
Category:
answer to question
reply to objection
correction
validation or rederivation

We thank the reviewer for their positive feedback. We include a point-by-point answer to their report in attachment.

Attachment:

reply_to_report1.pdf

---

## Round 1 · Referee Report · Anonymous (Referee 2) · 2024-9-20

Report

Please see the attached report. I would recommend the paper for publication if some concerns are addressed in a revision.

Attachment

Recommendation

Ask for minor revision

  • validity: -
  • significance: -
  • originality: -
  • clarity: -
  • formatting: -
  • grammar: -

Author:  Mauro Pastore  on 2024-11-07  [id 4945]

(in reply to Report 2 on 2024-09-20)
Category:
answer to question
reply to objection
correction
validation or rederivation

We thank the reviewer for their positive feedback. We provide a thorough reply to their report in attachment.

Attachment:

reply_to_report2.pdf

---

## Round 2 · Referee Report · Anonymous (Referee 1) · 2024-11-13

Strengths

I am re-listing all the strengths of this work that I have already underlined in my report for the first version of this paper:

- The authors provide a tight asymptotic characterization of the learning of Random Features Models (RFM) on a random polynomial target function, in various data/width/dimension regimes.

-They outline and identify data (resp. width) limited regimes where the RFM reduces to a kernel (resp. polynomial regression) method, as well as a non-trivial width~data regime, exhibiting in particular an interpolation peak phenomenon.

-The derivation relies on the replica method from statistical physics and several random matrix theory arguments and approximations. All steps are rather clearly justified, motivated, and discussed.

- The analytical findings are supported by convincing numerics.

- The consequences/takeaways of the analytical results are discussed, notably in terms of overfitting and expressive power.

Weaknesses

The authors clearly addressed all comments and concerns I had regarding the first version of the work. As far as I can see, they have also included additional discussions in accordance in the revised manuscript. I find the discussion and exposition of the results clear in the current version, and I do not have further concerns to report.

Report

The authors provided satisfactory clarifications to my questions regarding the first version of this work, and augmented the discussions in the manuscript in accordance. I thank the authors for this, and am in favor of acceptance of this version of the manuscript.

Recommendation

Publish (meets expectations and criteria for this Journal)

---

## Round 2 · Referee Report · Anonymous (Referee 2) · 2024-11-29

Report

I want to apologize for the delay in my review, and to thank the authors for taking well into account my comments. I read in detail through their answer and the revised parts of the manuscript, and all of my questions, remarks and criticisms have been addressed in detail. I am happy to recommend this revised version for publication.

Recommendation

Publish (meets expectations and criteria for this Journal)

---

## Editorial Decision

published